# Altered chromatin topologies caused by balanced chromosomal translocation lead to central iris hypoplasia

Wenmin Sun[1,3], Dan Xiong[2,3], Jiamin Ouyang[1,3], Xueshan Xiao[1,3], Yi Jiang[1], Yingwei Wang[1], Shiqiang Li[1], Ziying Xie[2], Junwen Wang[1], Zhonghui Tang [2] ✉ & Qingjiong Zhang [1] ✉

Despite the advent of genomic sequencing, molecular diagnosis remains unsolved in approximately half of patients with Mendelian disorders, largely due to unclarified functions of noncoding regions and the difficulty in identifying complex structural variations. In this study, we map a unique form of central iris hypoplasia in a large family to 6q15-q23.3 and 18p11.31-q12.1 using a genome-wide linkage scan. Long-read sequencing reveals a balanced translocation t(6;18)(q22.31;p11.22) with intergenic breakpoints. By performing Hi-C on induced pluripotent stem cells from a patient, we identify two chromatin topologically associating domains spanning across the breakpoints. These alterations lead the ectopic chromatin interactions between *APCDD1* on chromosome 18 and enhancers on chromosome 6, resulting in upregulation of *APCDD1*. Notably, APCDD1 is specifically localized in the iris of human eyes. Our findings demonstrate that noncoding structural variations can lead to Mendelian diseases by disrupting the 3D genome structure and resulting in altered gene expression.

With the increasing clinical implementation of next-generation sequencing, the gene diagnostic yield for monogenic diseases has improved but still rarely exceeds 50%[1–3]. For the majority of unsolved patients, this may be explained by technical shortcomings and incomplete knowledge of the functional consequences of most variants[4], especially those located in the noncoding regions. Having a wide application in genetic testing, the short-read sequencing platform has the power to detect single-nucleotide variations and small insertion and deletion variants, which are predominant in 88.6% of patients with molecular diagnosis[5]. The contribution of structural variations in coding and noncoding regions is, in fact, likely to be underestimated, because short-read sequencing methods have limited power to resolve structural variations, especially complex structural variations including inversions and translocations. Although long-read sequencing technology improves the identification of structural

variations, the analytical challenge around variant interpretation persists, especially for balanced structural variations with noncoding breakpoints[6–8]. Noncoding regions, representing approximately 98% of the human genome, have recently attracted attention in relation to genetically unsolved patients with Mendelian disorders[9–11]. To date, a number of noncoding variants have been identified, most of which were recognized according to a physical location in or adjacent to the known disease-causing gene, such as intronic, regulatory, upstream or downstream regions of known genes. The successful identifications of these noncoding variants are generally based on a targeted analysis around known genes, especially a common approach of whole-genome sequencing combined with RNA sequencing. These noncoding variants are reasonably linked with the aberrant expression of their adjacent known genes[12,13]. However, for variants at intergenic regions, evidence, both at the genetic level and in functional analysis, is

[1]State Key Laboratory of Ophthalmology, Zhongshan Ophthalmic Center, Sun Yat-sen University, Guangdong Provincial Key Laboratory of Ophthalmology and Visual Science, Guangzhou 510060, China. [2]Zhongshan School of Medicine, Sun Yat-sen University, Guangzhou 510080, China. [3]These authors contributed equally: Wenmin Sun, Dan Xiong, Jiamin Ouyang, Xueshan Xiao. ✉e-mail: tangzhh99@mail.sysu.edu.cn; zhangqji@mail.sysu.edu.cn

necessary to confirm the pathogenicity of these variants. To demonstrate this, we describe herein the identification of the genetic basis involving structural variations at the noncoding region in a large family with a unique type of iris developmental defect, i.e., central iris hypoplasia, which affects only the pupillary zone of the iris in both eyes.

## Results

### Identification of central iris hypoplasia in a large family

The unique condition in the large family (#71342) showed an autosomal dominant pattern of transmission (Fig. 1a). A total of 12 members of the family, including six affected and six unaffected individuals, participated in this study. The proband was referred to our clinic, Zhongshan Ophthalmic Center (Guangzhou, China), at one year of age because of photophobia since birth. The birth history and developmental milestones were unremarkable. A central absence of the iris

pupillary zone alone was observed in both eyes, with a pseudo-enlarged pupil of approximately 6 × 6 mm. The patient had a normal fovea but without nystagmus and other obvious abnormality according to her medical records. On a follow-up visit at six years of age, her visual acuity was 0.6 for both eyes, while her pupil size was enlarged to 8 × 8 mm owing to shrinkage of the peripheral iris. The remaining five affected members also had a history of photophobia since birth and a similar defect of the iris (Fig. 1b and Supplementary Fig. 1a, b). All six affected members had clear cornea and normal-like fundus with foveal reflex (Fig. 1b–e and Supplementary Fig. 1a–h) as normal controls (Fig. 1f–i). The axial length of the six affected individuals ranged from 19.17 mm to 25.32 mm. The corneal horizontal diameters, central corneal thickness, intraocular pressure, and central anterior chamber depth were available from three patients (IV:6, V:4, and V:6) and all of these parameters are in the normal range (Supplementary Table 1). Ultrasound biomicroscopy was performed on one patient (V:4), in

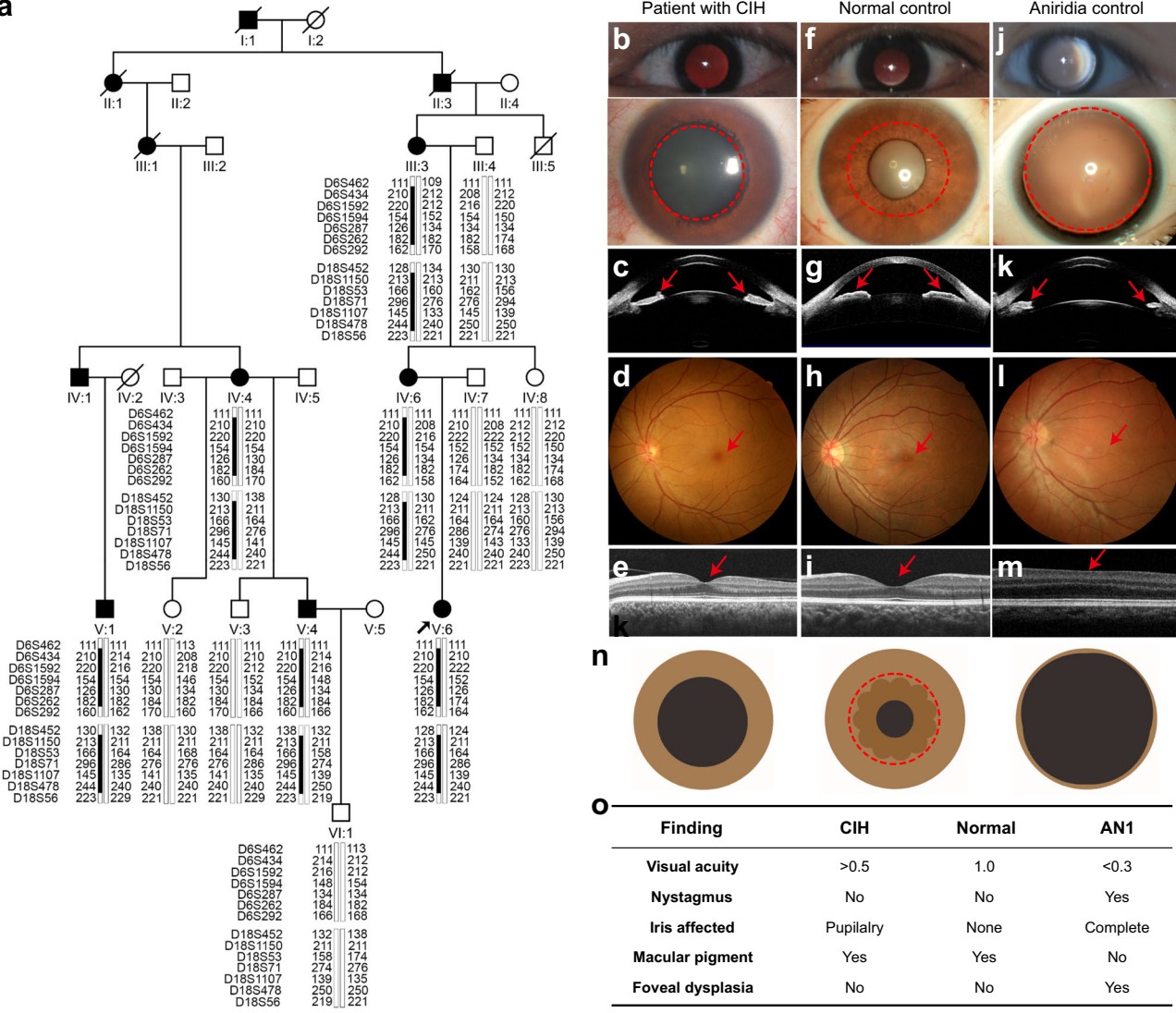

**Fig. 1 | The central iris hypoplasia phenotype maps to chromosome 6q15-q23.3 and chromosome 18.p11.31-q12.1. a** Pedigree of the family (#71342) with autosomal dominant transmitted central iris hypoplasia. The haplotype analysis of seven microsatellite markers adjacent 6q15-q23.3 and seven markers adjacent 18.p11.31-q12.1 are shown. Blackened bars indicate the risk haplotype co-segregating with central iris hypoplasia in all affected individuals in the family. Clinical features of central iris hypoplasia, namely a central hypoplasia involving the iris pupillary zone (**b**, **c**) with normal macular pigment in fundus photograph

(**d**) and normal foveal pit by OCT scan (**e**). **f–i** Normal features of the iris, fundus, and fovea. Characterizations of aniridia, including iris coloboma involving both pupillary and peripheral zones (**j**, **k**), absence of macular pigment (**l**), and foveal hypoplasia (**m**). **n** Schematic diagrams of iris features in central iris hypoplasia, normal control, and aniridia control, respectively. **o** Summary of differences among central iris hypoplasia, normal control, and aniridia control. CIH, central iris hypoplasia; OCT, optical coherence tomography; AN1, aniridia.

which the anterior chamber angle is open (Fig. 1c and Supplementary Fig. 1i, j). Gonioscopy examination of this patient suggested a mild malformation of the anterior chamber angle, that is an increasing presence of pectinate ligament (Supplementary Fig. 1k, l). The corneal endothelial cell density of the patient V:4 was 3003 cells/mm$^2$ with hexagonal cells of 68% in the right eye and 2892 cells/mm$^2$ with hexagonal cells of 73% in the left eye (Supplementary Fig. 1m, n). Two of the six, III:3 and IV:4, had age-related cataract on examination at ages of 72 and 55 years old, respectively (Supplementary Table 1). The central iris hypoplasia in this family is different from aniridia, which is characterized by iris coloboma involving both pupillary and peripheral zones, foveal hypoplasia, congenital nystagmus, and low visual acuity (Fig. 1j–o).

## A balanced translocation was identified based on mapped loci

A genome-wide linkage scan was performed on genomic DNA from all 12 individuals, and two-point linkage analysis yielded logarithm of the odds (LOD) scores greater than 2.0, with markers corresponding to chromosomes 6 and 18. Fine mapping and haplotype analysis confirmed the two loci on chromosomes 6q15-q23.3 and 18p11.31-q12.1 (Supplementary Table 2 and Fig. 1a). The locus on chromosome 6 mapped to a 38.0 cM region between D6S462 and D6S292, with D6S287 showing a maximum LOD score of 2.65 at theta = 0, while the other locus on chromosome 18 maps to a 38.0 cM region between D18S452 and D18S56 with D18S71 showing a maximum LOD score of 2.78 at theta = 0, theoretically the maximum scores for such a family.

To explore the pathogenic variation for the disease, whole-exome sequencing (WES) and/or whole-genome sequencing (WGS) analyses were performed on four affected patients, i.e., III:3, IV:4, V:4, and V:6, and two unaffected individuals, i.e., V:2 and V:3, based on a short-read platform. Both approaches failed to identify any causative single nucleotide variation (SNV) or insertion and deletion (InDel) inside the linkage intervals. To explore the potential structural variants, long-read genome sequencing was performed in an affected individual (V:4) through the Nanopore platform, on which a previously uncharacterized balanced translocation t(6;18)(q22.31;p11.22) (Fig. 2a) was identified with breakpoints of chr6:121619304 and chr18:10432739, respectively. Furthermore, fluorescent in situ hybridization (FISH) and karyotype analysis in an affected patient, i.e., IV:6, confirmed this translocation variation (Fig. 2b, c). Both breakpoints of the variation, which were at intergenic regions, were validated in all the six affected individuals but absent in all the six unaffected members by Sanger sequencing (Fig. 2d and Supplementary Fig. 2), a complete co-segregation with central iris hypoplasia within the family. The identification as well as the resolution of the translocation in members of the family are summarized in the Supplementary Table 3. The LOD score based on the haplotype encompassing the breakpoints was 3.08 at theta = 0, not only further supporting it as the disease-causing change for this family but also satisfactorily explaining the linkage to two different loci.

In addition to the large family described above, four additional unrelated families with central iris hypoplasia affecting the pupillary zone alone were identified in our clinic. Short-read WES and/or WGS on the four probands from these families excluded the presence of pathogenic variants in known genes of iris developmental defects. Karyotype records were available from two probands (#713867 and #71864), of whole one (#713867) had a de novo pericentric inversion of chromosome 18, inv(18)(p11.2 q11.2) (Supplementary Fig. 3), whereas the other (#71864) had a normal karyotype.

## Hi-C analysis revealed rewiring of the 3D genome by the translocation

To further explore the molecular mechanism underlying central iris hypoplasia caused by the balanced structural variation, peripheral blood mononuclear cells from one patient (IV:6) and one unaffected individual (IV:8) of the initial large family were reprogrammed to generate induced pluripotent stem cells (iPSCs). We performed parallel Hi-C sequencing, CUT&Tag, and RNA-seq analyses on the iPSCs samples from the two individuals. The Hi-C contact map revealed an aberrant interaction between chromosome 6 and 18 in the patient sample, confirming the presence of the translocation in the patient but not in the normal control (Supplementary Fig. 4). In the unaffected individual, the two breakpoints of the translocation were found to reside within two chromatin topologically associating domains (TADs) on chromosome 6 and 18, respectively (Supplementary Fig. 5). Notably, in the TADs on chromosomes 6 and 18 of the unaffected individual, we detected clusters of enhancers using H3K27ac CUT&Tag. Furthermore, analysis of H3K27ac profiles across various tissues from the EpiMap project (http://compbio.mit.edu/epimap/, accessed on May 15, 2023) revealed that the enhancer cluster on chromosome 6 exhibited tissue-specific activity (Supplementary Fig. 6), whereas the enhancer clusters on chromosome 18 showed ubiquitous presence in various tissues (Supplementary Fig. 7). A particularly strong activity of the enhancer cluster on chromosome 6 was present in the eye, retina, and embryonic stem cell, but not in immune cells, spleen, or liver (Supplementary Fig. 6). This enhancer cluster predominantly interacted with GJA1 promoter, which was located approximately 400 kb away within the TAD region of the unaffected individual. However, in the patient sample, the translocation disrupted the two TADs, leading to the formation of chromatin domains or neo-TADs over the breakpoints on the derivative chromosomes (Fig. 3a–c). To investigate the binding profile of chromatin structural factors at the neo-TAD boundaries, we retrieved the binding profiles of two key factors, CCCTC-Binding Factor (CTCF) and Cohesin (indicated by its subunit RAD21), from the EpiMap database (accessed on May 15, 2023). CTCF and RAD21 were strongly enriched at the boundaries of the neo-TADs, coinciding with their formation. Most importantly, the translocation juxtaposed the enhancer cluster mentioned above next to the APCDD1 promoter in a neo-TAD, leading to ectopic interactions between the enhancer cluster with APCDD1 promoter (Fig. 3a–c). By RNA-seq, the expression changes of genes in the vicinity of the two breakpoints were quantitated in iPSCs from the patient with central iris hypoplasia. The quantifications at individual replicate level of all the 12 genes close to the two breakpoints have been showed in the Supplementary Fig. 8 with the expression changes of all 40 genes within 2.5 Mb of each breakpoint in the Supplementary Data 1 (the full RNA-seq dataset in Supplementary Data 2). The results showed that APCCD1 was the only gene with a statistically significant expression change as defined by an adjusted $P < 0.05$ and a $|\log_2(\text{fold change})| > 1$ (Fig. 3d), which we confirmed by quantitative real-time PCR and Simple Western analysis (Fig. 3e–g). Moreover, six additional genes, including TBC1D32, GJA1, FABP7, TRDN, NDUFV2, and PIEZO2, show expression changes, especially the reduction of GJA1 RNA (FC = 0.57), which has the smallest FDR of 1.58E-93 (Supplementary Data 1). These findings suggest that the translocation rewired the three-dimensional chromatin structure, leading to an ectopia of the tissue specific enhancers and thereby dysregulating gene expression.

## The overexpression of APCDD1 was partially rescued by knock-out an enhancer region in iPSC from the patient

To validate the correlation of the ectopic enhancer cluster and APCDD1 overexpression, an 8.8 kb enhancer region around the first H3K27ac peak was targeted to be knocked out in iPSCs based on the cells from the patient with translocation (Fig. 4a). One clone of iPSCs with doubly heterozygous knockout of 10.6 kb with the endpoints varying by 1 bp was obtained (Fig. 4b, c). RT-qPCR results show that overexpression of APCDD1 RNA was partially rescued in iPSCs with enhancer knockout comparing with that in iPSCs with the translocation but without enhancer knockout (Fig. 4d). This result supports the potential

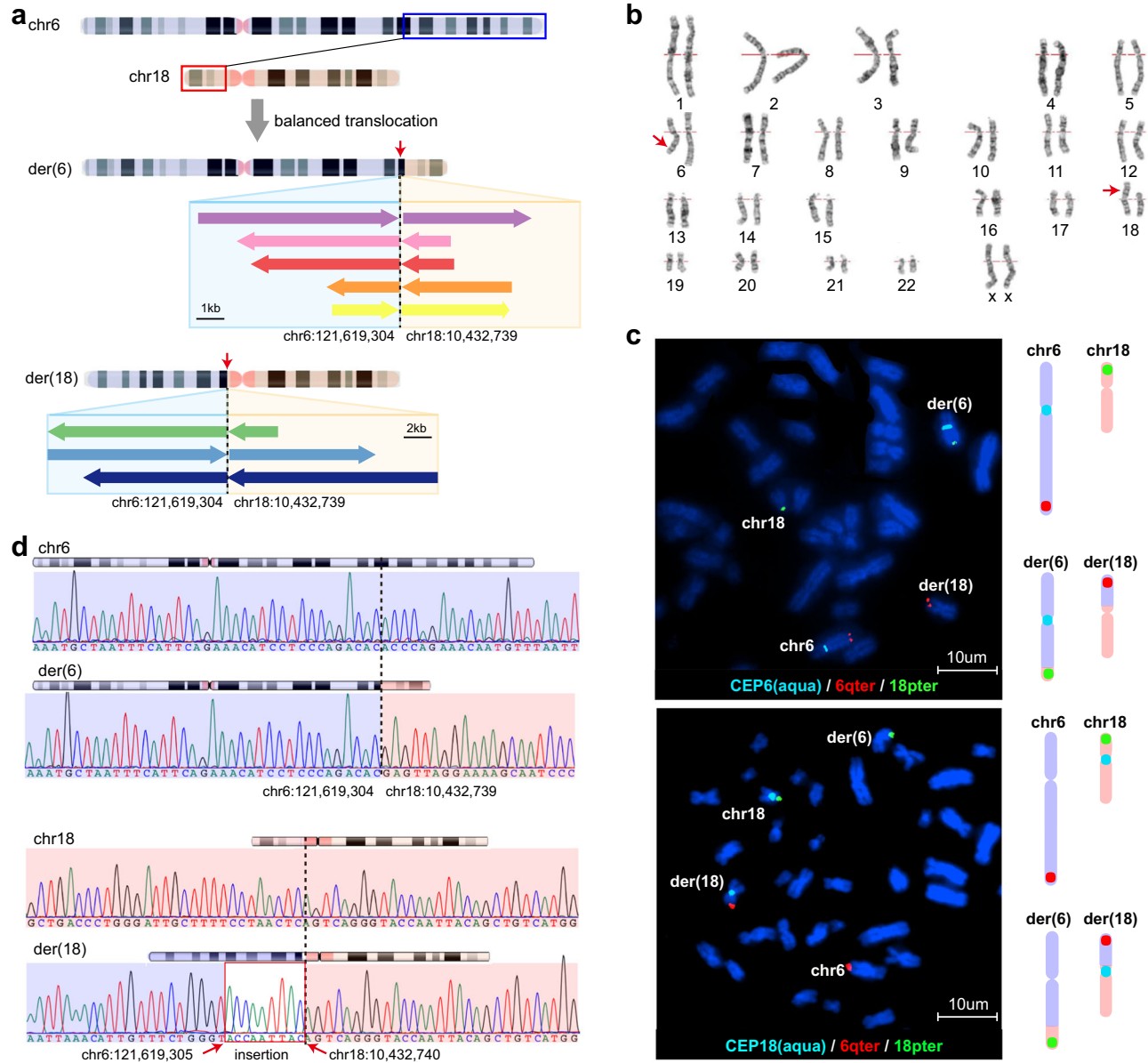

**Fig. 2 | A balanced translocation t(6;18)(q22.31;p11.22) with noncoding breakpoints in patients of the family (#71342). a** A balanced translocation t(6;18)(q22.31;p11.22) is identified by the Nanopore platform in an affected individual (V:4) of the family. Each colored box with an arrow indicates a long read that covers the breakpoint. The direction of the arrow indicates the direction of the reads aligned to the genome. Arrows pointing to the right represent reads aligned to the sense strand, while arrows pointing to the left represent reads aligned to the antisense strand. **b** Karyotype analysis of the affected individual (IV:6). A balanced translocation, namely 46,XX,t(6;18)(q22.2;p11.2), is identified (indicated by red arrows). **c** Fluorescent in situ hybridization of the affected individual (IV:6). In the upper panel, aqua signals labeled centromere of chromosome 6, red fluorescent signals indicate 6qter at normal chromosome 6 and derivate chromosome 18, whereas green fluorescent signals indicate 18pter at normal chromosome 18 and derivate chromosome 6. In the lower panel, aqua signals labeled centromere of the chromosome 18, red fluorescent signals indicate 6qter at the normal chromosome 6 and derivate chromosome 18, whereas green fluorescent signals indicate 18pter at the normal chromosome 18 and derivate chromosome 6. The experiment was twice repeated independently with similar results. A diagram showing normal and structurally variant chromosomes labeled by probes in each panel was provided on the right for clarity. **d** Sequence chromatograms of two breakpoints of the heterozygous translocation. The upper portion is normal sequences harboring breakpoints in chromosome 6 and chromosome 18, respectively. The lower portion is variant sequences flanking breakpoints of the translocation in the derived chromosomes. The positions of the breakpoints are indicated below the derived sequences.

mechanism that the ectopic chromatin interactions between *APCDD1* and enhancers results in the upregulation of *APCDD1*.

### APCDD1 was specifically expressed in the iris of human eyes and in the ciliary margin zone of embryonic mouse eyes

To assess the potential roles of APCDD1 in the iris, immunofluorescence staining was conducted in human eyes, where APCDD1 was specifically expressed in the anterior border layer as well as the stroma of the iris with co-staining with podoplanin (PDPN, a fibroblast marker) but separated from anti-actin, α-smooth muscle-FITC (α-SMA) (Fig. 5a). The specific expression of APCDD1 in the iris supported its role in iris structural maintenance by the location of iris fibroblasts. To further investigate APCDD1 expression during the embryonic development of the iris, immunofluorescence of Apcdd1 in embryonic mouse eyes at different stages, namely embryonic day 11.5 (E11.5), E13.5, E14.5, and E17.5, was performed. Apcdd1 was expressed in the

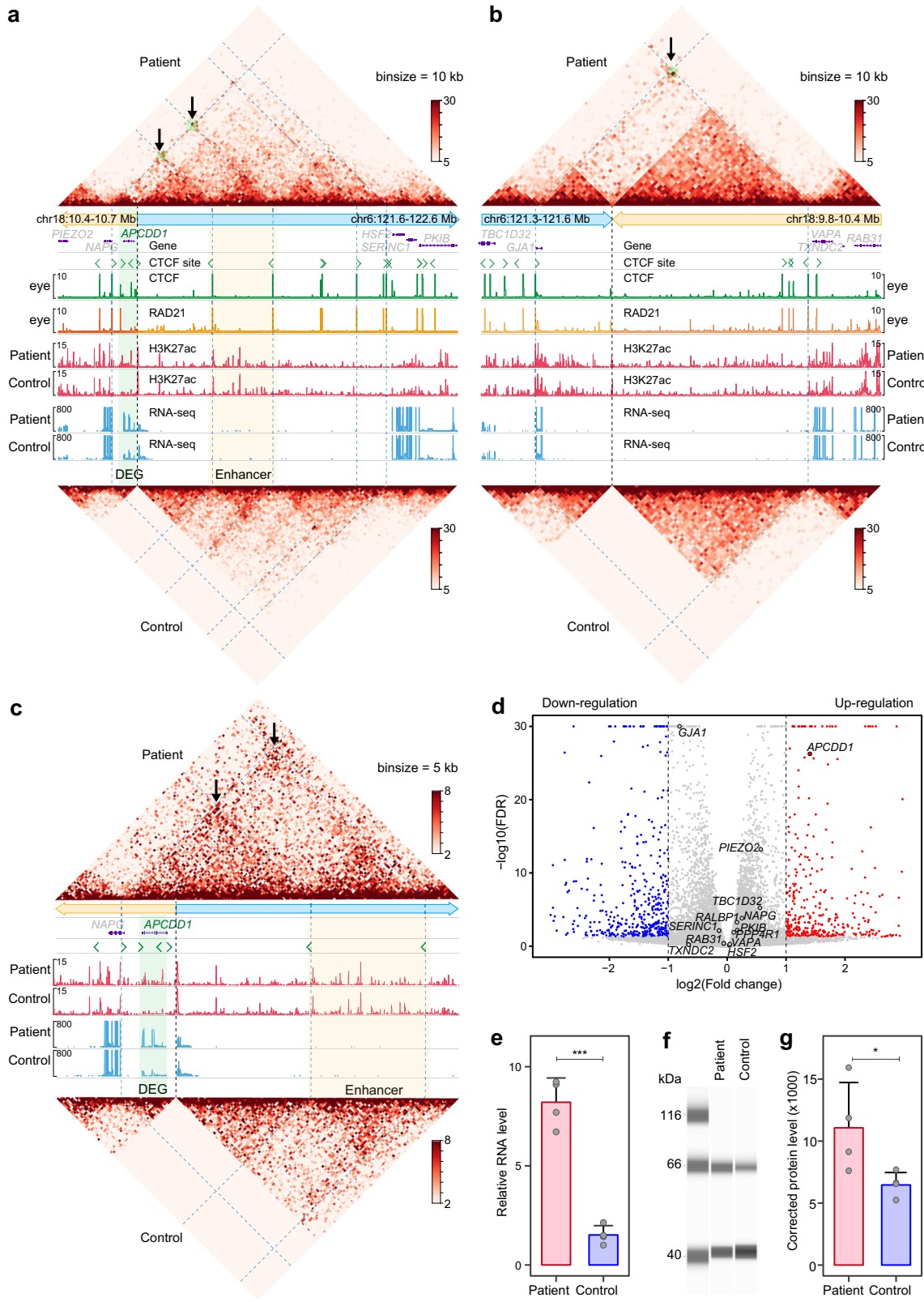

peripheral inner layer of the neural retina from E11.5 to E14.5 (Fig. 5b), which was partly co-staining with Cdo, a marker of the ciliary margin zone (Supplementary Fig. 9a), but not co-staining with βIII-Tubulin, which labeled the retinal neurons (Supplementary Fig. 9b). At E17.5, Apcdd1 was confined to the boundary of the ciliary body and the retina, which was preserved after birth (Fig. 5b and Supplementary Fig. 10). These results support that APCDD1 plays a role in the human

iris, whereas it may function in the development of the ciliary margin zone in mouse.

**Coloboma was present in *apcdd1l*-overexpressed zebrafish**
To validate the association between iris phenotypes and upregulation of *APCDD1*, mRNA of two zebrafish paralogues of human *APCDD1*, *apcdd1-like* (*apcdd1l*) and *LOC110438155*, was synthesized and injected

**Fig. 3 | Neo-TAD constrains tissue-specific enhancers to promote *APCDD1* expression. a, b** Two neo-TADs are formed on derivative chromosomes in patient samples. The top and bottom heatmaps display the interaction frequency between genomic regions surrounding the breakpoints of the translocation in iPSCs from the patient and normal control, with binning at a resolution of 10-kb. Red colors indicate higher interaction frequency. The black arrows and green dots indicate the aberrant interactions across breakpoint in the patient. The blue dash lines highlight the interactions across breakpoints in patient sample. The black dash lines indicate the locations of the two breakpoints. CTCF peaks and RAD21 peaks are strongly enriched at the boundaries of the neo-TADs. The orange box highlights an enhancer cluster (Enhancer), which is driven by the translocation next to the *APCDD1* promoter in a neo-TAD. The green box highlights a differentially expressed gene (DEG). **c** A zoom in a neo-TAD on the derivate chromosome 18 in (**a**). **d** Volcano plot shows differentially expressed genes between the patient and normal control. Each point represents a gene with the X-axis of log₂ fold change in gene expression and the Y-axis of statistical significance of the difference. Red and blue dots represent up-regulated and down-regulated genes, respectively. Gray dots indicate no significantly DEGs. The dashed lines indicate the threshold for significance (|log₂(Fold change)| ≥ 1). Notably, the *APCDD1* gene is up-regulation in patient sample. **e** Error-bar plot reveals a significant upregulation of *APCDD1* mRNA in patient compared with control by real-time PCR validation. Data are presented as mean values ± standard deviation and points indicate 4 replicates. One-side Student's *t* test was employed for pairwise comparisons. \*\*\**P* = 0.00031. **f** Simple western analysis indicates an upregulation of APCDD1 protein in the patient compared with control. Uncropped gel are shown in Source Data. **g** Error-bar plot reveals a significant upregulation of APCDD1 protein in patient compared with control by simple western analysis. Data are presented as mean values ± standard deviation and points represent 4 replicates. One-side Student's *t* test was employed for pairwise comparisons. \**P* = 0.041. Source data are provided as a Source Data file.

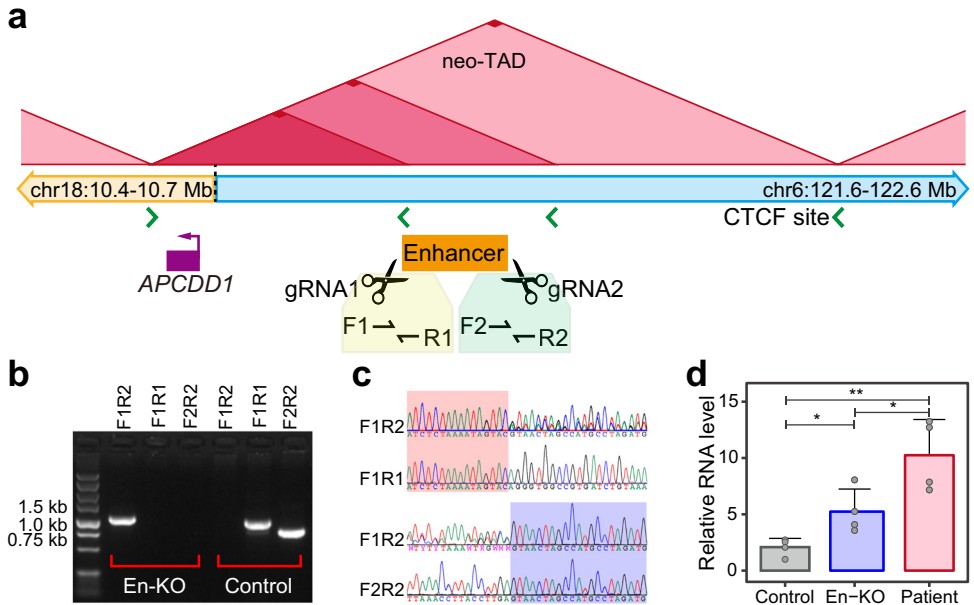

**Fig. 4 | The overexpression of *APCDD1* was partially rescued by knockout an enhancer region in iPSC from the patient. a** An 8.8 kb enhancer region around the first H3K27ac peak was targeted by designing a pair of gRNAs. Four primers (F1, R1, F2, and R2) were designed for amplification of two wild-type fragments covering the two sgRNAs and a knockout fragment covering the breakpoint. **b** Amplification of the knockout fragment with primer pairs of F1 and R2 in the knockout iPSCs and the normal fragments harboring the breakpoints of the knockout fragment with primer pairs of F1R1 and F2R2 in control iPSCs. Uncropped gel are shown in Source Data. **c** Sequence chromatograms of fragments amplified from different sets of primers demonstrated the boundaries of the 10.6 kb doubly heterozygous knock-out with the endpoints varying by 1 bp. **d** Error-bar plot reveals that *APCDD1* RNA was significantly decreased in iPSCs with enhancer knockout comparing with that in iPSCs with the translocation but without enhancer knockout. Data are presented as mean values ± standard deviation and points indicate 4 replicates. One-side Student's *t* test was employed for pairwise comparisons. En-KO, iPSCs with enhancer knockout. \**P* = 0.022. \*\**P* = 0.0059. Source data are provided as a Source Data file.

into zebrafish eggs at the one-cell stage. A phenotype of ocular coloboma was observed in 41.6% of *apcdd1l* overexpressed larvae and 55.1% of *LOC110438155* overexpressed larvae at day 3 post-fertilization (3dpf), which were significantly higher than that in wild-type (wt) larvae as well as in larvae injected with standard control mRNA (std) (Supplementary Fig. 11). By immunostaining with anti-phalloidin, a clear separation of the basement membrane at the optic fissure region was marked in the two groups of larvae with *apcdd1* overexpression, whereas the basement membrane was continuous in wt and std larvae at 3dpf (Supplementary Fig. 11). The coloboma due to failure of optic fissure closure observed in *apcdd1l* overexpressed larvae confirms the role of *APCDD1* in development of the iris.

## Discussion

In this study, a unique phenotype, namely central iris hypoplasia, was identified as an autosomal dominant trait in a large family. A whole-genome linkage scan, whole-genome sequencing, Hi-C, CUT&Tag, and RNA-seq analyses show that the molecular basis of the phenotype in this family is a balanced translocation t(6;18)(q22.31;p11.22) in a noncoding region that leads to alterations of the 3D genomic structure and causes aberrant interactions between enhancers and promoters of *APCDD1*, resulting in upregulation of the gene.

Taken together, the results of this study yield several implications in the area of medical genetics. First, the study provides evidence for the identification of complex structural variations, such as inversions and translocations, at noncoding intergenic regions relevant to Mendelian diseases. Although such complex structural variations can be identified by cytogenetic methods, they are rarely implicated in routine genetic diagnoses for Mendelian diseases[14]. Furthermore, complex structural variations are difficult to be detected by standard analysis of short-read sequencing platforms. The reasons include both the extreme diversity of complex structural variations making proper alignment difficult and the short length of each read creating limitations in the bioinformatic analysis[15]. The diversity of complex structural variations includes not only their types and sizes, ranging from 50 bp to several megabases, but also their frequent association with

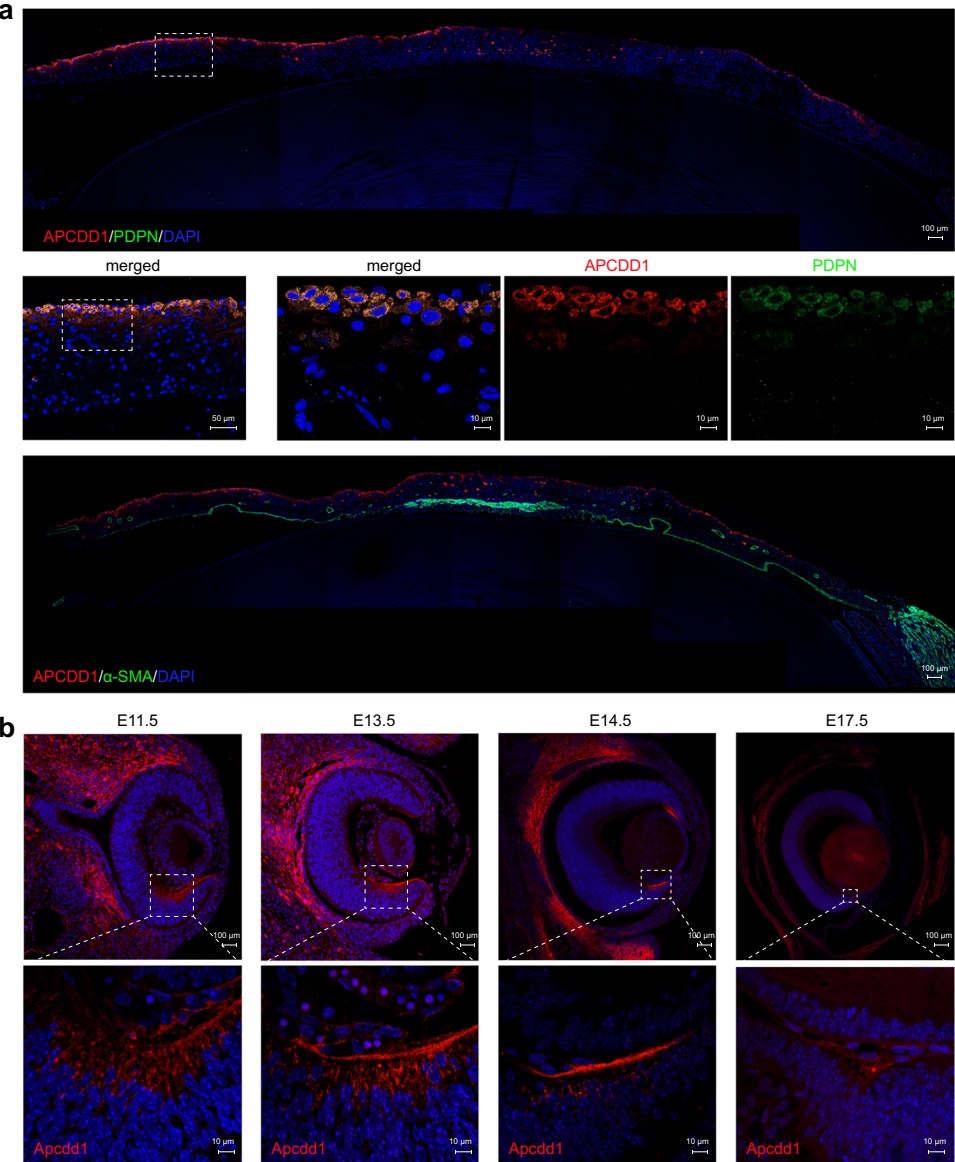

**Fig. 5 | Expression pattern of APCDD1 in the human iris and embryonic mouse eyes. a** Immunofluorescence in the human iris with anti-APCDD1. APCDD1 protein is expressed in the anterior surface of the human iris, which overlaps with that of podoplanin (PDPN) but is separate from that of α-SMA. Scale bars, 100 μm (10× magnification), 50 μm (20× magnification), 10 μm (63× magnification). **b** Immunofluorescence of Apcdd1 in embryonic mouse eyes at E11.5, E13.5, E14.5,

and E17.5, respectively. Apcdd1 is expressed in the peripheral inner layer of the neural retina from E11.5 to E14.5, and is confined to the boundary of the ciliary body and the retina at E17.5. Scale bars, 100 μm (whole eye) and 10 μm (63× magnification). These experiments were three times repeated indenpendently with similar results.

repeated DNA elements[16]. The extensive size and repetitive sequences of complex structural variations make it challenge to align the short sequencing reads to the reference genome as well as to evaluate the reliability of all identified complex structural variations when using the short-read sequencing platform[17]. Therefore, most complex structural variations have been identified by post hoc computational inference, which is rarely used in routine analysis[18]. With the development of long-read sequencing methodology, cases of genetic diseases caused by these structural variations are being identified increasingly frequently[19,20], most often when they directly disrupt protein-coding genes[8,21]. However, the effects of structural variations with noncoding breakpoints are underappreciated because of the limitation on interpreting enigmatic noncoding regions of the genome, especially in isolated cases where a causative correlation between structural variations and the phenotype are difficult to recognize. A uniform phenotype in large families with definite mapped loci based on genome-wide

linkage analysis provides unique opportunities to explore noncoding structural variations for Mendelian diseases, as shown by the current study's analysis of the large family with central iris hypoplasia.

In addition to the original large family, four additional families with central iris hypoplasia were identified in this study, although the genetic basis was only identified in one proband, namely a de novo balanced inversion inv(18)(p11.2q11.2). It is noteworthy that the inversion also occurred at chr18p11.2, the same location as the translocation seen in the original family. Therefore, the potential mechanism of the central iris hypoplasia in the patient with inversion variation may also be related to upregulation of *APCDD1*. However, the pathogenic mechanisms in the remaining three families have not been delineated and identification of the precise genetic mechanisms responsible for the disease in these families will be a subject for future studies.

Second, pathogenicity of noncoding structural variations is achieved through disrupting the three-dimensional chromatin

structure. This configuration is crucial for the regulation of at least some genes[22,23]. Technological advances in chromosome conformation capture, such as Hi-C analysis, have revealed the association between chromatin structure alterations and human diseases, including a few Mendelian diseases[22,24]. In this study, the structural variations occurred 200 kb downstream of *GJA1* and 20 kb upstream of *APCDD1*. In wild-type individuals, the tissue-specific enhancer clusters upstream of *GJA1* are constrained to regulate the expression of *GJA1* on chr6. However, in this family, the enhancer is driven by the three-dimensional chromatin rewiring to interact aberrantly with the promoter of *APCDD1*, leading to a significant upregulation of *APCDD1* expression and an approximately half downregulation of *GJA1* expression. These results provide an avenue to interpret the function and pathogenicity of noncoding variants, such as the balanced translocation with noncoding breakpoints in this study.

However, it is important to note that neo-TADs are not always functionally relevant with regard to phenotypes. Previous studies have been demonstrated that TADs may not themselves have strong regulatory effects on gene function but provide a structural framework for the functional regulation of the genes[25,26], and the results of this study also support this. It is not surprising that there are two neo-TADs spanning the two breakpoints of the translocation because of the proximity bias of Hi-C. Of the two neo-TADs resulting from the translocation in this patient, only the neo-TAD that is confirmed to upregulate *APCDD1* expression, was judged relevant to the iris hypoplasia, whereas the other TAD did not alter gene expression and thus might not be relevant to the iris hypoplasia phenotype. *GJA1*, which is known to cause eye anomalies, was also located in the vicinity of the breakpoint. The transcription level of *GJA1* showed a downregulation to approximately half in iPSCs from the patient with central iris hypoplasia compared with that in the wild type control individual with an FDR of 1.58E-93. While variants in *GJA1* are known to cause syndromes that include eye anomalies such as oculodentodigital dysplasia, to date these are either heterozygous missense variants implying a gain of deleterious function or a dominant-negative effect or homozygous null mutations. However, 50% decrease may accompany the heterozygous null allele that is well tolerated in human beings since individuals carrying heterozygous null allele are unaffected according to a systematic review in our previous study[27]. In addition to *GJA1*, several adjacent genes, such as *TBC1D32*, *FABP7*, *TRDN*, *NDUFV2*, and *PIEZO2*, also had expression dysregulation with significant FDR according to RNA-seq under lower threshold. There currently is no strong evidence suggesting that a 50% reduction of *GJA1* or the other adjacent genes is associated with a human phenotype. However, while our data suggest that upregulation of *APCDD1* is likely a primary mechanism responsible for the phenotype, possible contributions from expression changes in adjacent genes, either from a single gene (e.g., *GJA1*) or the combined effects of several genes cannot be excluded. In this regard, it is interesting to note that WNT signaling causes the *GJA1* product, Cx43, to translocate to the nucleus similar to its action on β-catenin, with which Cx43 interacts directly, suggesting that it might participate in the β-catenin destruction complex or perhaps serve as a co-transcriptional activator[28].

Third, the discovery of disease-associated genes acting through increased dosage effects has been particularly challenging, as in the case of *APCDD1* in this study. The majority of genes causing human diseases are identified as a result of loss-of-function variants and occasionally via gain-of-function[29,30]. However, variants with increased dosage effects are easily missed by current variant prioritization strategies in clinical genetic pipelines[31], especially variants in noncoding regions including structural variations. Additional studies aimed at understanding the upregulation of functional genes through coding or noncoding elements, as in the current study, may improve our ability to predict the effect of variants that regulate genes contributing to human diseases, which may in turn elucidate the genetic cause of

diseases in the approximately half of unsolved patients with inherited disease[30,32,33]. Finally, the only additional phenotype reported to result from mutations in *APCDD1* is hypotrichosis simplex without ophthalmic disease, which is associated with a Leu9Arg variant[34]. This is interesting because variants in *LSS* were initially reported to cause cataracts in the eye but were subsequently identified as a frequent cause of hypotrichosis[35,36].

*APCDD1* was initially isolated in 2002 as a target of the beta-catenin/T-cell factor 4 complex[37]. Subsequent studies found that it functions as an inhibitor of Wnt signaling and further as a dual inhibitor of both Wnt and BMP signaling in the developing nervous system and skin[34,38]. Moreover, *Apcdd1* has been reported to coordinate vascular development of retinal blood vessels by modulating Wnt/Norrin signaling activity in the mouse[39]. These results suggest that *APCDD1* has complicated functions central to animal development and human disease. It is important to note that the phenotype in this family appears to result from upregulation of *APCDD1* in patients with central iris hypoplasia, which is contrary to the dominant negative pathogenic mechanism of the Leu9Arg variant seen in previous studies[34,38]. These results suggest that *APCDD1* has a complex function, both in the iris and in the hair follicle, and must be precisely regulated for proper development.

Moreover, it has previously been shown that APCDD1 can interact in vitro with WNT3A and LRP5, both of which are essential components of Wnt signaling[34]. However, neither LRP5 nor WNT3A were detected in the human iris in a previous study on cell atlas generation of the human ocular anterior segment[40]. At least 625 proteins have been implicated in the Wnt signaling pathway (Uniprot, https://www.uniprot.org/), of which 136 coding genes are expressed in fibroblasts of the human iris, including APCDD1. By RNA-seq analysis five of the 136 genes show a significantly altered expression in iPSCs in the patient as compared to the control, including *APCDD1*, *GREM1*, *RSPO2*, *WIF1*, and *WLS*. Moreover, it is noted that variants in *WLS*, encoding wntless, have been reported to be associated with iris coloboma[41]. This suggests that APCDD1 and WLS might possibly interact or share a similar pathway, although this needs to be validated in further studies.

As both gene expression and enhancer activity may be specific for individual tissues, a major limitation of this study is the unavailability of iris tissue from the affected patients, which would allow evaluation of gene expression and enhancer activity so that these are necessarily based on iPSC samples. Although the iPSCs provide information for human tissues, including eyes, at early stages of development, a better disease model may be generating iris organoids based on iPSCs. However, this approach is not yet technically feasible. In addition, it is challenging to recapitulate the central iris hypoplasia phenotype in a zebrafish model with overexpression of *APCDD1* paralogues because of the difficulty of achievement of temporal and spatially restricted *apcdd1* overexpression in the zebrafish iris. In this study, mRNAs of *APCDD1* paralogues were microinjected in one-cell eggs of zebrafish to generated models with overexpression of the gene. However, the coloboma phenotype in larvae with *apcdd1* overexpression is different from the human phenotype although it suggests a role in development of the iris. Even if feasible, this might not recapitulate the human phenotype due to the two paralogues in the zebrafish, differences in developmental signaling pathways, and 450 million years of evolutionary separation.

In summary, identification of noncoding structural variations affecting gene expression and discovery of the iris-associated phenotype in this study demonstrate the potential role of noncoding structural variations in hereditary diseases, which has yet to be fully explored or understood. Hopefully this approach will help elucidate how other hereditary diseases might be caused by noncoding structural variants in further studies focusing on the mechanisms through which alterations of three-dimensional chromatin structures affect gene expression.

## Methods

### Ethics statement

Clinical data and peripheral blood were collected from probands with central iris hypoplasia and their available family members under approval by the Institutional Review Board of the Zhongshan Ophthalmic Center, Sun Yat-sen University (2011KYNL012). Written informed consent in accordance with the tenets of the Declaration of Helsinki was obtained from all participants or their guardians prior to this study. All participants or their guardians consented to the use of their samples and medical information to publish. The authors affirm that human research participants (or their parents/guardians) provided informed consent for publication of the images in Fig. 1 and Supplementary Fig. 1, and for publication of the information in Supplementary Table 1. The generation of iPSCs from peripheral blood of participants received ethics approval from the Institutional Review Board of the Zhongshan Ophthalmic Center, Sun Yat-sen University (2011KYNL012). The use of postmortem human ocular tissues, which were obtained from the Eye Bank of Guangdong Province, was approved by the Institutional Review Board of the Zhongshan Ophthalmic Center, Sun Yat-sen University (2023KYPJ200). All animal experiments were performed according to the Association for Research in Vision and Ophthalmology (ARVO) Statement for the Use of Animals in Ophthalmic and Vision Research and guidelines established by the Animal Experimental Ethics Committee of Zhongshan Ophthalmic Center, Sun Yat-sen University (W2021005-1 for mouse experiments and 2020-177 for zebrafish experiments).

### Subjects

Five probands with central iris hypoplasia involving pupillary zone and their available family members, including a four-generation family (#71342), were identified from Zhongshan Ophthalmic Center, Guangzhou, China. Available clinical data and peripheral blood were collected from the participants. Sex and/or gender of participants was not considered in the study design, where sex and/or gender of participants was determined based on self-report.

### Genome-wide linkage analysis

Twelve individuals of the large family #71342, including six patients and six unaffected members, participated in this study. Genomic DNA was isolated from peripheral blood samples of the 12 individuals using a phenol-Chloroform DNA isolation method[42]. Genome-wide linkage analysis was carried out on genomic DNA from the 12 members as follows: PCR amplification of 382 highly polymorphic 5′-fluorescently labeled microsatellite markers was conducted using the ABI PRISM Linkage Mapping Sets Version 2 according to the manufacturer's specifications (Applied Biosystems, Foster City, CA). Alleles were separated using an ABI 3100 DNA analyzer and then assigned using GeneMapper (Applied Biosystems, Foster City, CA). Two-point linkage analysis was conducted with the MLINK program of the FASTLINK implementation of the LINKAGE program package[43,44]. The iris hypoplasia in the family was analyzed as an autosomal dominant trait with complete penetrance and a disease allele frequency of 0.0001. For fine mapping around candidate loci, additional markers were genotyped using an M13-tailed primer PCR method[45].

### Short read high-throughput sequencing

Short read high-throughput sequencing was conducted on genomic DNA samples from the family#71342 through Illumina sequencing platforms (Illumina, San Diego, CA)[46], of which WES on four affected individuals (III:3, IV:4, V:4, V:6) and one unaffected control (V:3) and WGS on two affected (V:4 and V:6) and one unaffected (V:2). Exome was captured using a SureSelect Human All Exome kit (Agilent, Santa Clara, CA) for WES and exome-enriched DNA fragments were sequenced on the illumine Hiseq system (Illumina, San Diego, CA) with a depth of at least 125-fold. The library of WGS was prepared using a

trueSeq DNA Sample preparation Kit (Illumina, San Diego, CA) and was sequenced on the library was qualified and then sequenced on the Illumina HiSeq genome analyzer platform (Illumina, San Diego, CA) with an average coverage of 30-fold.

### WES and WGS data analyses

The reads were aligned with the consensus sequence (UCSC hg19) for variant detection by the Burrows–Wheeler Aligner (BWA v0.7.15)[47]. Variants, including single-nucleotide variants (SNVs), small insertions and deletions (InDels), and structural variations, were filtered using GATK (v3.8.0)[48] and annotated using ANNOVAR[49]. Variants interpretation in candidate loci were conducted using multistep bioinformatics analysis as follows: (1) excluding non-coding variants as well as synonymous variants without predicted splicing effect; (2) excluding variants with minor allele frequency (MAF) ≥ 0.1%; (3) excluding missense variants predicted as tolerate; (4) excluding variants without segregation with diseases in the family.

### Long-read whole-genome sequencing

Long-read genome sequencing was performed on the genomic DNA of an affected individual (V:4) via Oxford Nanopore platform as used before[46]. The procedures of Nanopore sequencing included qualification of DNA samples, library preparation, and sequencing. After the qualification, large-size fragments were selected from native genomic DNA of each proband using Blue Pippin. The library was prepared by end repair, A-tailing, and ligation of sequencing adapters on the fragments. Finally, the prepared library was sequenced using PromethION sequencing with the total bases of at least 45 Gb with an average depth of 15-fold.

### Long-read whole-genome sequencing data analysis

The base calling was carried out on the data generated from PromethION using the Guppy software and then low-quality reads with a mean qScore of less than seven were excluded. Reads were aligned with human reference genome (hg38) using NGMLR (v0.2.7)[50] and alignments were visualized in IGV. Structural variations were detected using Sniffles and annotated using ANNOVAR[49]. The frequencies of structural variations were annotated based on the Database of Genomic Variants (DGV, http://dgv.tcag.ca/dgv/app/home), Decipher (https://www.deciphergenomics.org/#) and 1000 Genomes (http://phase3browser.1000genomes.org/index.html) databases if an overlapping region accounted for at least 50% of the structural variations from database and the patient.

Potentially pathogenic structural variations within the linked region were identified as follows: (1) low-confidence structural variations with coverage of no more than four reads were excluded; (2) the remaining high-confidence structural variations would be excluded if they were present in DGV or 1000 Genomes databases; (3) Sanger sequencing was conducted for breakpoint validation and segregation of candidate structural variations using primers as listed in Supplementary Table 4.

### Karyotype and Fluorescent in situ hybridization

Conventional cytogenetic analyses, including karyotype and fluorescent in situ hybridization (FISH), were conducted on peripheral blood lymphocytes from the affected individual (IV:6) by commercial services from AmCare Genomics Lab (Guangzhou, China) and Guangdong Youning Biological Technology Company (Guangzhou, China). Karyotype was performed using the G-banding technique with a 550-band resolution. FISH was performed using four probes with three colors, namely a chromosome 6 centromere probe (aqua), a chromosome 18 centromere probe (aqua), a chr6qter probe (red), and a chr18pter probe (green) according to the results of long-read WGS. Twenty metaphases were analyzed and the description of results were made according to the International System for Human Cytogenetic Nomenclature (2016).

### Reprogramming peripheral blood mononuclear cells into iPSCs

Lymphocytes were isolated from peripheral blood samples of the affected IV:6 and her unaffected sibling IV:8 through density gradient centrifugation via lymphocyte separation medium (MP Biomedicals, China). Peripheral blood mononuclear cells (PBMCs) were cultured using StemPro™−34 complete medium (Thermo Fisher Scientific, MA, USA) following manufacturer instructions. PBMCs of the two individuals were reprogramed into iPSCs using the CytoTune -iPS 2.0 Sendai Reprogramming Kit (Thermo Fisher Scientific, MA, USA) according to the manufacturer instruction. Colonies with iPSCs-like morphology were manually picked and transferred onto vitronectin-coated culture plates using Essential 8™ Medium (Thermo Fisher Scientific, MA, USA).

### Hi-C sequencing

The in situ Hi-C experiments of at least 5 million iPSCs per sample from the patient and the control was conducted by a commercial service from Annoroad Gene Technology (Beijing, China)[51]. Briefly, genomic DNA of iPSCs was crosslinked in situ using serum-free DMEM with 2% formaldehyde and then were flash-frozen in liquid nitrogen before digested. The crosslinked cells were lysed and digested with MboI restriction enzyme. After biotinylated and ligated, the fragments were purified and assessed for quality. The qualified samples were reverse crosslinked and then submitted for standard library constructions, including sonication fragment, end repair, A-tailing, biotin enrichment, adapter ligation, and amplification. Two libraries per sample were sequenced on the Illumina NovaSeq platform in PE150 mode with total data of at least 400 Gb per sample.

### Hi-C data analysis

Hi-C data of iPSC from the patient and the normal control, with two biological replicates each, was processed using HiC-Pro pipline (v2.11.1)[52]. Paired-end reads were aligned to the human reference genome (hg38) downloaded from the Ensembl genome browser using bowtie2 with parameters '--very-sensitive -L 30 --score-min L,−0.6,−0.2 --end-to-end --reorder' for global alignment and '--very-sensitive -L 20 --score-min L,−0.6,−0.2 --end-to-end --reorder' for local alignment. Mapped reads were filtered to retain only those with a mapping quality greater than 10 and paired by using 'mergeSAM.py' script in HiC-Pro pipeline. We then extracted the fragments involved in chromatin interactions from the mapped reads using 'mapped_2hic_fragments.py' script in HiC-Pro pipeline. For the patient and the normal control samples, we obtained 716,025,801 and 707,547,068 validPairs for further processing, respectively. These validPairs were used to construct the contact maps at 500-kb and 10-kb resolution using the 'build_matrix' function of HiC-Pro pipeline with parameters '--matrix-format upper --binsize 500,000' and '--matrix-format upper --binsize 10,000', respectively. To remove systematic biases and artifacts, the contact maps were subjected to normalization using iterative correction and eigenvector decomposition (ICE) method[53]. The normalized contact maps were converted to h5 format using 'hicConvertFormat' function of HiCExplorer (v3.5.1)[54]. Visualizing the 500-kb resolution whole-genome contact map revealed a significant interaction signal between chromosome 6 and chromosome 18 (Supplementary Fig. 4), which suggested the presence of chromosomal translocation events involving these two chromosomes. To provide a more detailed representation of the chromatin interactions at the breakpoints of the translocation, we employed 'hicPlotMatrix' function of HiCExplorer (v3.5.1) to visualize the chromatin interactions surrounding the breakpoints, with binning at a resolution of 10-kb. The contact matrix revealed that the breakpoints of translocation were situated within the two active TADs (topologically associating domains), as shown in Supplementary Fig. 5. However, in the patient, the translocation led to disruption of those two TADs. On the derivative chromosomes forming the neo-TAD, there were strong chromatin interactions between enhancers and promoters across the breakpoints, as depicted in Fig. 3a, b. To investigate the binding profile of chromatin structural factors at the neo-TAD boundaries, we retrieved the binding profiles of two key factors, CTCF and RAD21, from the EpiMap database (http://compbio.mit.edu/epimap/)[55]. The EpiMap database provides comprehensive chromatin-state maps and genomic annotations for various human tissues. The retrieved data revealed robust signals of CTCF and RAD21 binding at the neo-TAD boundaries (Fig. 3a, b).

### RNA-seq

Approximately one million iPSCs per sample were immersed in RNAiso Plus solution (Takara Bio Inc., Japan) and submitted to Annoroad Gene Technology (Beijing, China) for RNA sequencing. Total RNA was extracted and quality and quantity were detected using Nanodrop 2000 (Thermo Fisher Scientific, MA, USA) as well as Agilent 2100 Bioanalyzer (Agilent Technologies, CA, USA). The mRNA was enriched from qualified samples using Oligo (dT) coated beads and then was fragmented using fragmentation buffer. The RNA fragments were reverse transcribed to synthesize cDNA using random primers. The library was prepared through sequentially end repair, A-Tailing, adapter ligation, size selection, and amplification. Finally, the quantified libraries were sequenced on the Illumina HiSeq platform in PE150 mode with total data of at least 10 Gb per sample.

### RNA-seq data analysis

RNA-seq data of iPSC from the patient and the normal control, with five and four biological replicates respectively, were aligned to human reference genome (hg38) using hisat2 (v2.1.0)[56]. We removed the low-quality mapping reads (MAPQ < 20) using 'samtools view' function of samtools (v1.3.1)[57]. The remaining reads were sorted using 'samtools sort' function of samtools. The coverage of sorted reads across the genome was calculated using 'bamCoverage' function of deepTools (v.3.4.3)[54] with the parameter '--binsize 10'. To quantify the gene transcription levels, we counted reads on exons of each gene using 'htseq-count' function of HTSeq (v0.11.2)[58], utilizing human reference gene annotations download from Ensembl genome browser. To exclude the impact of low or non-expressed genes on the analysis of differential expressed genes, we removed genes with cumulative reads count from all replicates less than 20, resulting in a set of 23,541 genes for analysis. Differential expressed analysis was performed using DEseq2 (1.30.1)[59] R package with thresholds '|log$_2$(fold change)| > 1 and FDR < 0.05'. We identified 372 up-regulation genes and 454 down-regulation genes. A volcano plot illustrating the differentially expressed genes were generated using custom R code and ggplot2 (v3.3.3) package. The differential expression analysis revealed a significant up-regulation of *APCDD1* in patient samples, with a two-fold increase compared to normal controls (as shown in the volcano plot). On the other hand, we observed a down-regulation trend of *GJA1* in patient samples (Fig. 3d and Supplementary Fig. 8).

### Cleavage Under Targets and Tagmentation (CUT&Tag)

CUT&Tag library construction[60,61] was performed as follows: Approximately 0.5 million iPSCs per sample from the patient and control were resuspended and attached to Concanavalin A beads (Bangs Laboratories, IN, USA). The iPSCs on ConA beads were incubated at RT with primary antibody against H3K27ac (#39133, 1:50 dilution, Active motif) for one hour, and then with secondary antibody (#SAB3700894, 1:50 dilution, Sigma-Aldrich) for one hour. After washing cells attached on ConA beads three times, cells were then incubated for one hour with pA-Tn5 (Vazyme, China) pre-loaded with annealed adapter complex. After rinsing cells attached on ConA beads three times, cells were resuspended and incubated for one hour to allow DNA tagmentation. After quenching the fragmentation, the cells were added with proteinase K and then incubated for one hour. Post-incubation, a phenol–chloroform–isoamyl alcohol mix was added to the solution and vortexed at full speed to mix. The solution was

transferred to a phase lock tube and centrifuged. The supernatant was transferred to a fresh tube and DNA was precipitated via the addition of ethanol. The samples were incubated at −80 °C freezer for 30 min and subsequently centrifuged for 20 min. The supernatant was carefully discarded, and the pellet was air dried for 10 min and resuspended in a desired amount of EB buffer. DNA concentration was estimated using the Qubit. PCR amplification was performed using universal N5 primer and indexed N7 primer and PCR cycle number was optimized. Amplified products were purified using 1: 1.3X Ampure bead. Finally, eluted library DNA were sent for sequencing using illumina Nova platform.

## Cut&Tag data analysis
Cut&Tag data of iPSC from the patient and the normal control, with two biological replicates each, were aligned to the human reference genome (hg38) using 'bwa mem' of BWA (v0.7.15)[47] with default parameters. We filtered out reads with low mapping quality (MAPQ < 20) reads using 'samtools view' function of samtools (v1.3.1) and sorted the remaining reads using 'samtools sort' function. PCR duplicates were subsequently removed using 'MarkDuplicates' function of Picard. The coverage of uniquely mapped reads across the genome was calculated using 'bamCoverage' function of deepTools (v.3.4.3) with the parameter '--binsize 10'. H3K27ac modification peaks were identified using 'macs2 callpeak' function of MACS2 (v2.2.7.1)[62] with default parameters. In iPSC from the patient and the normal control, 43,558 and 52,769 peaks were detected, respectively. We then assessed the reproducibility between the biological replicates and found that the Pearson correlation coefficients were greater than 0.91, indicating a high level of reproducibility. H3K27ac always serves as a marker for enhancer activity. We found an enhancer cluster downstream of GJA1, consisting of multiple H3K27ac peaks, which was present in both patient and the normal control. Analysis of the contact map of iPSC from the patient revealed was found that this enhancer cluster interacts with the promoter of APCDD1. To demonstrate the tissue specific activity of the enhancer cluster, we downloaded 16 H3K27ac signals from EpiMap database for various tissues, including embryonic eye, embryonic eye retina, eye retinoblastoma (WERI-Rb-1 cell line) immune cells, spleen, liver, etc. These H3K27ac profiles revealed strong enhancer activity in the eye, retina, and embryonic stem cells, but no enhancer activity in immune cells, spleen, liver, etc. (Supplementary Figs. 6,7).

## Enhancer-knockout in the iPSC with the translocation
An enhancer-knockout iPSC was generated based on the iPSC from the patient with translocation by a commercial service from the Cyagen Biosciences (Suzhou, China). Briefly, three pairs of single-guide RNAs (sgRNAs) were designed targeting an 8.8 kb region of the enhancer cluster (Chr6:121846253-121855088). After editing efficiency test, the most efficient pair of sgRNAs (the sequences are ATCTCTAAAATAGTA CAGGGTGG and TAGGCATGGCTAGTTACTCAAGG, respectively) was selected for enhancer knockout in the iPSC from the patient with translocation. For enhancer knockout, $1 \times 10^6$ iPSCs were electroporated with Cas9 protein and sgRNA and plated on five Vitronectin-coated 96-well plates with one-cell per well. After 72 h, wells with cells were maintained by changing the medium once a week. The single cell-derived cells per well were passaged as two copies at approximately 80% confluence, of which one copy was maintained for genotyping. For genotyping, four primers (Supplementary Table 4) were designed for amplification of two wild-type fragments covering the two sgRNAs and a knockout fragment covering the breakpoint. Direct amplification was performed on cells using TransDirect Animal Tissue PCR Kit (TransGen, China) according to the manufacturer's instruction.

## Human ocular tissues
Postmortem human eyeballs were obtained from the Eye Bank of Guangdong Province, which were from an eye donor who died of meningioma. Written informed consent consistent with the tenets of the Declaration of Helsinki was obtained from the donor prior to obtaining the eyeballs.

## Quantitative real-time PCR
Total RNA samples were prepared from iPSCs and zebrafish larvae using a Takara RNAiso Plus (TaKaRa Bio Inc., Japan) and were reverse transcribed using a PrimerScript™ RT Reagent kit (TaKaRa Bio Inc., Japan) according to the vendor's protocol. Quantitative real-time PCR (qPCR) was performed to measure the mRNA expression levels of genes of interest using specific primers (Supplementary Table 4) in a QuantStudio Dx (ThermoFisher, MA, USA). *GAPDH* for human and actin for zebrafish were used as the normalization control and each reaction was run in triplicate.

## Simple western analysis
Protein was extracted from approximately 0.5 million iPSCs per sample, which were rinsed with ice-cold DPBS and then lysed in RIPA buffer. After centrifugation at $13,400 \times g$ for 20 min at 4 °C, the lysates were collected from the resultant supernatants. The protein concentrations of resultant lysates were measured by a PierceTM BCA Protein Assay kit. The Simple Western analysis, or capillary electrophoresis immunoblotting, were performed using the ProteinSimple Wes Simple Western system with a 12–230 kDa Master Kit (Proteinsimple, Santa Clara, CA), which is a next-generation capillary immunoassay platform for automated protein separation and quantitation[63,64]. Briefly, protein samples were diluted with 0.1 × Sample buffer to achieve a concentration of 1.5 µg/µl. Samples, namely 5 µL mixtures of 5 × Master mix and protein dilutions, and primary antibodies as well as other reagents were loaded to the WES setup for the Simple Wes assays using 12–230 kDa separation capillaries, according to the manufacturer's protocol (Proteinsimple, Santa Clara, CA). The following primary antibodies were used: anti-APCDD1 (#PA5-98605, 1:200 dilution, Thermo Fisher Scientific) and anti-GAPDH (#2118S, clone 14C10, 1:200 dilution, Cell Signaling Technology) (Supplementary Table 5). The separation electrophoresis and immunodetection steps automatically run in the capillary system. The Campass software 6.0 (Protein Simple) was used to analyse the digital image and quantify data of the detected protein (Supplementary Data 3). The quantity of APCDD1 protein was calculated by correction of the housekeeping GAPDH protein loading to the same sample.

## Animal maintenance
Mice used in this study have a congenic C57BL/6J background and were derived by backcrossing to a parental inbred strain for at least ten generations. They were bred and housed in a specific pathogen-free mouse facility with a regular 12 h light and 12 h dark cycle and persistent environment temperature ranging between 20 °C and 22 °C with 40–60% humidity. To obtain time-matched embryonic mice, male and female adult mice were caged together in the afternoon. Vaginal plugs were checked in the next morning. Noon on the day a virginal plug was found was regarded as embryonic day 0.5 (E0.5). Eyes from wild-type mice (C57BL/6J) with different developmental stages, including embryonic stages E11.5, E13.5, E14.5, E17.5, and postnatal stages P1, P14, 1 M, 2 M, were used for immunofluorescence experiment. Sex was not considered in study design because it would not cause protein staining differences in immunofluorescence experiments. Wild-type zebrafish (AB Danio rerio) embryos were acquired from the Zebrafish Facility, Laboratory Animal Center, Sun Yat-sen University. The embryos were bred by natural spawning and maintained at 28.5 °C in a constant temperature with a 14 h light: 10 h dark cycle. Zebrafish embryos and larvae from wild-type AB strains were used for microinjections and immunofluorescence experiments. Sex was not considered in study design, as it is not determined at the early developmental stages studied in this study.

## In vitro synthesis of mRNAs of *APCDD1* paralogues and microinjections

The plasmid was synthesized by Tsingke Biotechnology Co., Ltd. (Beijing, China), in which the *apcdd1l* and *LOC110438155* (the two zebrafish paralogues of human *APCDD1* obtained from the National Center for Biotechnology Information, https://www.ncbi.nlm.nih.gov/) cDNA sequence was cloned into the pCS2+ vector. The plasmid with *apcdd1l* (pCS2 + -*apcdd1l*) and that with *LOC110438155* (pCS2 + -*LOC110438155*) as well as pCS2+ vector were digested with NotI at 37 °C. The mRNA, including *apcdd1l*, *LOC110438155*, and pCS2 + , were synthesized sing the mMESSAGE mMACHINE transcription kit (Thermo Fisher Scientific, MA, USA) according to the manufacturer's instructions. An equal dose (200 pg) of mRNA from pCS2 + -*apcdd1l*, pCS2 + - *LOC110438155*, and pCS2+ (as standard control) was injected into one-cell-stage embryos following the standard protocol[65], respectively. Embryo development was evaluated at 3dpf. At 3dpf, the larvae were anesthetized by 0.03% tricaine. The whole body and ocular at lateral side were captured by stereoscopic fluorescence microscope M205FA (Leica, Germany).

## Histology and immunofluorescence microscopy

Tissues from mice (after transcardiac perfusion with cold saline) and zebrafish larvae at 3dpf were fixed in 4% paraformaldehyde (PFA), whereas human eyeballs were immersed in FAS Eyeball Fixative Solution (Servicebio, China). Fixed human and mouse tissues were processed and paraffin-embedded microtome sections (at 4 µm thickness) were prepared by Servicebio (Wuhan, China). Fixed zebrafish larvae were embedded in OCT (4583, Scigen Scientific Gardina) and cut into 10-µm-thick frozen sections. For paraffin sections from human eyeballs and mouse eyes, deparaffinization and antigen retrieval were required. Briefly, sections were deparaffinized using xylene and were rehydrated using gradient alcohol, namely 100%, 95%, 75%, and 30% of alcohol, respectively. Rehydrated sections were then incubated in citrate buffer (10 mM pH 6.4) at 98 °C for 30 min for antigen retrieval. These sections as well as frozen sections from zebrafish larvae were blocked with 5% normal goat serum (NGS) (30 min at room temperature, RT) and then incubated with a specific primary antibody overnight at 4 °C. After washed by PBST, proper fluorescent secondary antibodies as well as DAPI were used for incubation for 1 h at RT and then treated with TrueBlack® Lipofuscin Autofluorescence Quencher. For double labeling, two primary antibodies from two different species were mixed together, and so were secondary antibodies. The following antibodies were used: anti-APCDD1 (#Bs-1565R, 1:200 dilution, Bioss), anti-CDO (#AF2429-SP, 1:200 dilution, R&D systems), anti-Podoplanin (#14-5381-82, clone eBio8.1.1 (8.1.1), 1:200 dilution, Thermo Fisher Scientific), anti-α-SMA (#F3777, clone 1A4, 1:500 dilution, Sigma-Aldrich), Anti-β-Tubulin III (#T8578, clone 2G10, 1:500 dilution, Sigma-Aldrich), Donkey Anti-Mouse IgG H&L (Alexa Fluor® 488) (#ab150105, 1:1000 dilution, Abcam), Donkey Anti-Rabbit IgG H&L (Alexa Fluor® 568) (#ab175470, 1:1000 dilution, Abcam), (Supplementary Table 5). Images of stained sections were captured using a Zeiss LSM980 (Carl Zeiss, Baden-wurberg, Germany) confocal microscope.

## Statistical analysis

The relative mRNA levels or the corrected protein levels between the patient and the normal control as well as the relative mRNA levels among the patient, the patient with enhancer knockout, and the normal control were statistically analyzed using a one-tailed unpaired Student's *t* test (*p* value indicated in the legend). The data are presented as mean ± SD by error-bar plot. The relative mRNA levels among four groups of larvae, namely *apcdd1l*-overexpressed larvae, *LOC110438155*-overexpressed larvae, standard control larvae, and wild-type larvae, were analyzed for statistical significance using a one-tailed unpaired Student's *t* test (*p* value indicated in the legend). The data are

presented as mean ± SD by error-bar plot. The observed coloboma in four groups of larvae was assessed for statistical significance using pearson's *Chi*-squared test (*p* value indicated in the legend). Statistical data, biological replicate numbers, and individual data points are listed in the Source data file.

## Reporting summary

Further information on research design is available in the Nature Portfolio Reporting Summary linked to this article.

## Data availability

The data from CUT&Tag, Hi-C, and RNA-Seq are generated in this study and have been deposited in the Gene Expression Omnibus (GEO) database under accession codes GSE266513, GSE266514, GSE266515 and the Genome Sequence Archive (GSA) database under accession code HRA004690, which can be publicly accessible. The data from long-read WGS generated in this study have been deposited in the GSA database under accession code can be HRA004690 and can be publicly accessible. The data used in the binding profile for CTCF and RAD21 are downloaded from the EpiMap database (https://epigenome. wustl.edu/epimap/data/imputed/) under the accession number BSS00329. The data used in the analysis for H3K27ac signals in 16 tissues are downloaded from the EpiMap database (https:// epigenome.wustl.edu/epimap/data/imputed/) under the accession number as follows: embryonic eye (BSS00329), embryonic eye retina (BSS01504), eye retinoblastoma (BSS01890), ESC (BSS00277), muscle (BSS01290), kidney (BSS01086), skin (BSS01587), frontal cortex (BSS00369), B cell (BSS00095), T cell (BSS01346), thymus (BSS01820), lung (BSS01140), spleen (BSS01630), liver (BSS01159), urinary (BSS01876), and pancreas (BSS01406). Source data are provided with this paper.

## Code availability

This paper does not report original codes. Any additional information required to reanalyze the data reported in this paper is available from the lead contact upon request.

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

## Acknowledgements

The authors thank all patients and family members for their participation. This work was supported by grants from National Natural Science Foundation of China (82171056), the Science and Technology Planning Projects of Guangzhou (202102010271 and SL2024A03J00525), and the Fundamental Research Funds of the State Key Laboratory of Ophthalmology.

## Author contributions

Q.Z. conceived the study. Q.Z. and Z.T. provided overall supervision of the study. Q.Z., X.X., S.L., W.S., Y.J., Y.W., and J.W. collected venous blood and clinical data and prepared the genomic DNA. XX and WS performed linkage analysis. X.X. and L.S. performed the short-read high throughput sequencing. W.S., X.X., and D.X. conducted long-read whole genome sequencing. W.S., J.O., and Y.J. cultured cells for Hi-C, RNA-seq, and CUT&Tag analyses. Z.X. prepared libraries of CUT&Tag. D.X. and Z.T. analysed data from Hi-C, RNA-seq, and CUT&Tag. W.S., J.O., and Y.W. performed animal experiments and immunofluorescence. W.S., D.X., J.O., and X.X. wrote the manuscript. Q.Z. and Z.T. provided critical revision of the manuscript. All authors reviewed, contributed to, and approved the manuscript.

## Competing interests

The authors declared no competing interests.
