## [Peer Review File · Nature Communications]

Altered chromatin topologies caused by balanced chromosomal translocation lead to central iris hypoplasiaREVIEWER COMMENTS

Reviewer #1 (Remarks to the Author):

This study focuses on understanding the molecular diagnosis of a particular form of circular iris coloboma, a developmental defect affecting the iris of the eye. Despite advances in genomic sequencing, many patients with Mendelian disorders remain undiagnosed due to challenges associated with noncoding regions and complex structural variations. The authors conducted a genome-wide linkage scan in a large family and identified two genomic regions, 6q15-q23.3 and 18p11.31-q12.1, associated with the condition. By employing long-read sequencing, the researchers discovered a balanced reciprocal translocation, $t(6;18)(q22.31;p11.22)$, with breakpoints located in intergenic regions. Further investigation using Hi-C analysis on induced pluripotent stem cells from an affected patient revealed two novel chromatin topologically associating domains spanning the translocation breakpoints. These altered chromatin topologies resulted in abnormal interactions between the promoter of a gene called APCDD1 on chromosome 18 and tissue-specific enhancers on chromosome 6. The consequence of this aberrant interaction was the upregulation of APCDD1 mRNA transcription and protein expression. The authors also show that APCDD1 is normally found in the iris of human eyes and in the ciliary margin zone of embryonic mouse eyes. The study provides further evidence that balanced rearrangements in noncoding regions of the human genome can lead to Mendelian diseases by disrupting the three-dimensional structure of the genome. Such disruptions can cause abnormal interactions between enhancers and promoters, resulting in altered gene expression and developmental defects like iris coloboma.

The study by Sun et al provides compelling clinical and molecular evidence for the pathogenicity of this unique translocation causing coloboma. The paper is well written, the experiments support the conclusions, and the authors use cutting edge technology. Therefore, the paper should be of interest to the readers of Nature Communications. However, I have several concerns with the manuscript in its current format:

Major

1: While the authors provide compelling clinical and human genetics evidence for the pathogenicity of their translocation, they do not provide final proof that enhancer hijacking is actually the disease mechanism in this particular case. In figure 3 they identify an enhancer cluster as the potential disease driver and show a specific overexpression of APCDD1. However, the paper would be much more convincing if the authors would demonstrate that the knockout of the enhancer sequences leads to a rescue of the overexpression of APCDD1 in the patient cell line.

2: Another option to prove the functional link between the enhancers and the gene would be to overexpress APCDD1 in the zebrafish under the control of the enhancers. However, this is probably technically not possible.

Minor:

1: The authors nicely show in figure 3 that the translocation results in two distinct neo TADs. However the authors should at least mention or discuss that neo TAD formation in itself is not surprising due to the proximity bias of HiC and might not be functionally relevant. The functional more relevant information here is the overexpression of APCDD1.

2: Could the authors comment on the fact why the translocation was not identified by short read based whole genome sequencing?

Reviewer #2 (Remarks to the Author):

Overall, the paper is an impressive collection of analysis to identify a translocation associated with a Mendelian disorder in a large family through both short and long read sequencing. Then moreover to functionally assess its impact through functional genomic approaches in iPSCs and imaging and perturbations in model organisms. The provides both an interesting observation on disease with an implied induction of a gene and an interesting case study to think about the impacts of translocation.

My main comments are that important details are sometimes hard to identify or given briefly, and further analysis or clarification of their current data is necessary to really describe the breakpoint and to be certain that APCDD1 is the primary gene implicated by their analysis.

- The text should be made more clear as to which individuals the translocation was identified and to what resolution. As far as I can tell the Nanopore and Sanger is V4, the karyotype and iPSC generation were V6, the main text is not especially clear on this. If this is correct, presumably the precise coordinates of the breakpoints identified in V4 can be validated with V6 WGS and Hi-C data. Ideally this validation should be included as supplement.

- A deeper survey of the detail of the authors data should be given surrounding the breakpoint to make it explicit that its APCDD1 impacted. Does GJA1 change in expression? This gene had been implicated in eye disease phenotypes and the authors should be very explicit about whether GJA1 changes in expression and whether that is relevant to the phenotype present. More generally, as the authors have gone to the excellent effort of generating Hi-C, RNA-seq and CUT&Tag in patient iPSCs, the authors should show:

* Quantifications at individual replicate level of APCDD1 and all genes in the vicinity of the breakpoints (at least those in Fig 3). Was APCDD1 the only gene with dysregulated expression?

* A zoom in of the Hi-C, RNA-seq, and CUT&Tag data surrounding their putative inducing enhancer and the breakpoints in patient and control iPSCs, realigning the data to the translocated sequence if necessary. The patient appears to have gained H3K27ac peaks, and the RNA-seq appears to show data spanning from APCDD1 to over the break point.

* The use of public data to evaluate the regulatory potential across different tissues to provide context is great, however, the main text should be made more explicit as to when public data is used. For example, when stated that CTCF and RAD21 were found at neo-TAD boundaries, it should be made clear that this is in public eye data and not patient iPSCs.

* The authors should supply the visualisation of public data at the chr18 locus too, are there eye specific enhancers implied to regulate APCDD1, what happens to these under the translocation.

- The authors should acknowledge the limitation that since they evaluate gene expression in pluripotent cells that the enhancer activity and gene expression changes may not precisely reflect those within the relevant cells within the iris.

- The discussion of the literature and role of APCCD1 is limited, and whilst it relates to the retina it feels necessary that the authors cite and consider:

Mazzoni J, Smith JR, Shahriar S, Cutforth T, Ceja B, Agalliu D. The Wnt Inhibitor Apccdd1 Coordinates Vascular Remodeling and Barrier Maturation of Retinal Blood Vessels. *Neuron*. 2017 Dec 6;96(5):1055-1069.e6. doi: 10.1016/j.neuron.2017.10.025. Epub 2017 Nov 16. PMID: 29154126; PMCID: PMC5728434.

This should be done considering the authors work and Shinomura et al, 2010. The section on WLS is speculative and unless the authors wish to provide further results or supplemental material to support this, the comment should be made in the discussion rather than appear as a result.

- Can the authors be more explicit on the significance of the additional families with CIC and how these relate to the implied up-regulation of APCDD1. Are these all relevant to be included in this study?

- The discussion should perhaps underscore more clearly that as APCDD1 is already expressed in the tissue of interest and expression is hypothesised to increase in patient with the translocation that CIC is not expected to result from loss-of-function of APCDD1.

Reviewer #3 (Remarks to the Author):

This well-written manuscript reports a new ocular phenotype attributable to a balanced reciprocal chromosomal translocation. Key strengths include the authors' convincing demonstration that the translocation generates new chromatin topologically associated domains, juxtaposing enhancer elements adjacent to the candidate causative gene. Fully validated, this finding would demonstrate a structural chromosomal variant in a non-exonic region causes a novel Mendelian disease. However, several areas require enhancement, with one additional experiment, to fully support the conclusions. I hope the following assist the authors with their revisions:

1) The phenotype reported is not of a coloboma; a group of disorders caused by failure of closure of the choroid fissure (please google 'coloboma image' and compare with Fig. 1b). Rather than implicating the ~35th gene in coloboma, the authors have in fact identified a more interesting phenotype, where most of the iris appears not to develop.

Apart from correcting the text, the authors need to illustrate the ocular anterior segment in greater detail, and at higher resolution for most/all the affected individuals. This would reveal any phenotypic variability. They could beneficially include multiple additional data [anterior chamber depth, central corneal thickness, endothelial cell density, axial length measurements, etc.]. The iris induces corneal development, so some of these parameters, and potentially the trabecular meshwork and or intraocular pressure may be altered. If UBM imaging is feasible, the manuscript would provide a very detailed first report of a novel phenotype.

2) The fluorescent in situ data, and most important part of Figure 2, is small and tough to read. Larger and enhanced resolution panels highlighting the normal autosome and both structurally variant chromosomes, would assist. These three should also appear as larger insets, ideally with similar color labelling to Fig 2a, to demonstrate each rearrangement. The authors could consider FISH with 3 or 4 differentially-labelled probes to both chromosome 6 and 18, to illustrate the rearrangements' complexity [currently separate images are shown for chr.6 and then chr. 18].

3) Analysis of the structural variant with high-throughput chromosome confirmation capture, and then CUT & Tag sequencing, was viewed as very strong. However, there appears a logic leap to APCDD1 being the causative gene/sole causative gene, based on its position and upregulated expression on RNA-sequencing and quantitative immunoblotting. The current volcano plot while nice, is not particularly relevant if over-expression of a wnt-bmp inhibitor dysregulates numerous downstream genes. Instead, we need to see whether other genes adjacent to the breakpoints are dysregulated, or if APCDD1 is the only candidate. This would be addressed by providing the full RNA-seq dataset [as a supplemental table], and a list of genes within 2.5Mb of each breakpoint with their degree of altered expression.

Once the above has been tackled, a key experiment is missing. The authors should demonstrate whether CRISPR interference targeting the enhancer elements identified via CUT & Tag, abrogates increased APCDD1 mRNA expression. Evidence of partial normalization of APCDD1 expression with CRISPRi / multiplexed CRISPRi would neatly validate the central mechanism proposed in this paper.

4) The zebrafish data are not compelling, and several concerns exist.

First, as frequently observed, there are two zebrafish paralogs of the mammalian ortholog. However,

this complexity is neither discussed, nor is the second ocular-expressed paralog investigated (only *apcdd1*-like).

Second, the logic of injecting mRNA of a dual *bmp* and *wnt* inhibitor into a one cell zebrafish embryo to model tissue and temporally-specific human APCDD1 over-expression, is unclear. The inhibitor's broad roles cause profound effects at picogram doses, that do not recapitulate the human phenotype [compare Figs. 1b and 4c].

Third, the observed delayed fusion of the zebrafish choroid fissure [a coloboma], reflects altered patterning and not a tissue-specific defect. BMP and Wnt signaling have essential roles establishing and maintaining the dorso-ventral axis of the retina/eye, so an inhibitor will alter levels of multiple *bmp/wnt* ligands, inducing colobomata without supporting the human phenotype.

Achieving temporal and spatially-restricted mRNA over-expression in the zebrafish iris would be challenging. Even if feasible, it may not recapitulate the human phenotype due to multiple paralogs, differences in developmental signaling pathways, and 450 million years of evolutionary separation. I will be interested in the other reviewers' comments on this point. Since I think it would be unreasonable to require *in vivo* electroporation in murine/xenopus, a better approach may be to discuss the challenges modelling a disorder caused by highly specific dysregulation of a single gene during a restricted temporal interval, with mRNA micro-injection of one cell zebrafish embryos. Fig 4 could then be relegated to supplemental, to demonstrate that this has been attempted, but proved unsuccessful.

Authors' Response to Comments from the Editors and the Reviewers

In the following we list the comments from the editor and reviewers in **bold** followed by our changes/explanations in plain text. Changes made in the manuscript are highlighted in blue.

REVIEWER COMMENTS

Reviewer #1 (Remarks to the Author):

This study focuses on understanding the molecular diagnosis of a particular form of circular iris coloboma, a developmental defect affecting the iris of the eye. Despite advances in genomic sequencing, many patients with Mendelian disorders remain undiagnosed due to challenges associated with noncoding regions and complex structural variations. The authors conducted a genome-wide linkage scan in a large family and identified two genomic regions, 6q15-q23.3 and 18p11.31-q12.1, associated with the condition. By employing long-read sequencing, the researchers discovered a balanced reciprocal translocation, t(6;18)(q22.31;p11.22), with breakpoints located in intergenic regions. Further investigation using Hi-C analysis on induced pluripotent stem cells from an affected patient revealed two novel chromatin topologically associating domains spanning the translocation breakpoints. These altered chromatin topologies resulted in abnormal interactions between the promoter of a gene called APCDD1 on chromosome 18 and tissue-specific enhancers on chromosome 6. The consequence of this aberrant interaction was the upregulation of APCDD1 mRNA transcription and protein expression. The authors also show that APCDD1 is normally found in the iris of human eyes and in the ciliary margin zone of embryonic mouse eyes. The study provides further evidence that balanced rearrangements in noncoding regions of the human genome can lead to Mendelian diseases by disrupting the three-dimensional structure of the genome. Such disruptions can cause abnormal interactions between enhancers and promoters, resulting in altered gene expression and developmental defects like iris coloboma.

The study by Sun et al provides compelling clinical and molecular evidence for the pathogenicity of this unique translocation causing coloboma. The paper is well written, the experiments support the conclusions, and the authors use cutting edge technology. Therefore, the paper should be of interest to the readers of Nature Communications. However, I have several concerns with the manuscript in its current format:

Major

1: While the authors provide compelling clinical and human genetics evidence for the pathogenicity of their translocation, they do not provide final proof that enhancer hijacking is actually the disease mechanism in this particular case. In figure 3 they identify an enhancer cluster as the potential disease driver and show a specific overexpression of APCDD1. However, the paper would be much more convincing if the authors would demonstrate that the knockout of the enhancer sequences leads to a rescue of the overexpression of APCDD1 in the patient cell line.

Our response: We appreciate that the reviewer's confirmation of our main finding and the valuable suggestion. Since it is difficult to knockout the whole 200 kb enhancer cluster, an 8.8 kb region around the first H3K27ac peak was targeted to knock out this region in iPSCs based on the cells from the patient with translocation (Fig. 4a). This is actually a doubly heterozygous knockout of 10.6 kb with the endpoints varying by 1 bp (Fig. 4b-c). RT-qPCR results show that overexpression of *APCDD1* RNA was partially rescued in iPSCs with enhancer knockout comparing with that in iPSCs with the translocation but without enhancer knockout (Fig. 4d). This result supports the potential mechanism that the ectopic chromatin interactions between *APCDD1* and enhancers results in the upregulation of *APCDD1*. These results have been added to the Results and Methods sections as well as the revised Fig. 4.

2: Another option to prove the functional link between the enhancers and the gene would be to overexpress APCDD1 in the zebrafish under the control of the enhancers. However, this is probably technically not possible.

Our response: We agree with the reviewer. As reviewer #3 has pointed out, there are conceptual and practical difficulties in investigating these enhancers in a zebrafish model. Because the sequence of the enhancer cluster is not conserved between human beings and zebrafish, the best approach would be to knock in the human enhancer sequence near to the target gene in zebrafish. However, there are two problems with this approach. First, it is technically difficulty to knock in a region as large as the ~200 kb enhancer, and whether the human enhancer elements would be functional in the zebrafish target gene is unknown. Therefore, we have generated an enhancer knockout iPS cell line to validate the association of the enhancer cluster and overexpression of *APCDD1* as suggested above.

Minor:

1: The authors nicely show in figure 3 that the translocation results in two distinct neo TADs. However the authors should at least mention or discuss that neo TAD formation in itself is not surprising due to the proximity bias of HiC and might not be functionally relevant. The functional more relevant information here is the overexpression of APCDD1.

Our response: Thank you very much for this insightful comment. It has been demonstrated that TADs may not themselves have strong regulatory effects on gene function but provide a structural framework for the functional regulation of the genes,^{1,2} and the results of this study also support this. We agree with the reviewer that it is not surprising that there are two neo TADs spanning the two breakpoints of the translocation because of the proximity bias of Hi-C. Although at least three genes were located in the two neo TADs, only *APCDD1* had significantly increased expression, suggesting that of the two neo TADs, only the one in which *APCDD1* was located, was considered to be functionally relevant. This have been added in the Discussion section:

“However, it is important to note that neo-TADs are not always functionally relevant with regard to phenotypes. Previous studies have been demonstrated that TADs may not themselves have strong regulatory effects on gene function but provide a structural framework for the functional regulation of the genes,^{28,29} and the results of this study also support this. It is not surprising that there are two neo TADs spanning the two breakpoints of the translocation because of the proximity bias of Hi-C. Of the two neo TADs resulting from the translocation in this patient, only the neoTAD that is confirmed to up-regulate *APCDD1* expression, was judged relevant to the iris aplasia, whereas the other TAD did not alter gene expression and thus might not be relevant to the iris aplasia phenotype.”

2: Could the authors comment on the fact why the translocation was not identified by short read based whole genome sequencing?

Our response: The translocation variation is a type of complex structural variations (cxSVs), which are difficult to be detect by standard analysis of short-read whole genomic sequencing. The reasons include both the extreme diversity of cxSVs making proper alignment difficult and the short length of each read creating limitations in the bioinformatic analysis.³ The diversity of cxSVs includes not only their types and sizes, ranging from 50 bp to over megabases, but also their frequent association with repeated DNA elements.⁴ The extensive size and repetitive sequences of cxSVs make it challenge to align the short sequencing reads to the reference genome as well as to evaluate the reliability of all identified cxSVs when using the short-read sequencing platform.⁵ Therefore, most cxSVs have been identified by *post hoc* computational inference, which is rarely used in routine analysis.⁶ This has been added in the Discussion section:

“Furthermore, complex structural variations are difficult to be detect by standard analysis of short-read sequencing platforms. The reasons include both the extreme diversity of complex structural variations making proper alignment difficult and the short length of each read creating limitations in the bioinformatic analysis.¹⁸ The diversity of complex structural variations includes not only their types and sizes, ranging from 50 bp to several megabases, but also their frequent association with repeated DNA elements.¹⁹ The extensive size and repetitive sequences of complex structural variations make it challenge to align the short sequencing reads to the reference genome as well as to evaluate the reliability of all identified complex structural variations when using the short-read sequencing platform.²⁰ Therefore, most complex structural variations have been identified by *post hoc* computational inference, which is rarely used in routine analysis.²¹”

Reviewer #2 (Remarks to the Author):

Overall, the paper is an impressive collection of analysis to identify a translation associated with a Mendelian disorder in a large family through both short and long read sequencing. Then moreover to functionally assess its impact through functional genomic approaches in iPSCs and imaging and perturbations in model organisms. The provides both an interesting observation on disease with an implied induction of a gene and an interesting case study to think about the impacts of translocation.

My main comments are that important details are sometimes hard to identify or given briefly, and further analysis or clarification of their current data is necessary to really describe the breakpoint and to be certain that APCDD1 is the primary gene implicated by their analysis.

- The text should be made more clear as to which individuals the translocation was identified and to what resolution. As far as I can tell the Nanopore and Sanger is V4, the karyotype and iPSC generation were V6, the main text is not especially clear on this. If this is correct, presumably the precise coordinates of the breakpoints identified in V4 can be validated with V6 WGS and Hi-C data. Ideally this validation should be included as supplement.

Our response: We thank the reviewer for these important comments. The identification as well as the resolution of the translocation in members of the family are now included as the Supplementary Table 3. The Sanger sequence chromatograms of the two breakpoints were detected in the six affected members and are now shown in Supplementary Fig. 2, whereas they were not seen in the six unaffected family members by PCR amplification. A Sanger sequence tracing of unaffected individual IV:8 is included as a negative control in Fig. 2.

Supplementary Table 3. The resolution of the translocation identified in members of the family.

Methodology	Individual ID	Resolution
Karyotype	IV:6	46,XX,t(6;18)(q22.2;p11.2)
FISH	IV:6	ish t(6;18)(6p+,18p+;6q+,18q+)
WES	III:3, IV:4, V:4, V:6	No
WGS	V:4, V:6	/
Nanopore	V:4	chr6:121619304 & chr18:10432739
Sanger	III:3, IV:4, IV:6, V:1, V:4, V:6	chr6:121619304 & chr18:10432739
Hi-C	IV:6	chr6:121619304 & chr18:10432739

Notes: FISH, fluorescent in situ hybridization. WES, whole-exome sequencing. WGS, whole-genome sequencing. /, The translocation was not identified by standard analysis of short-read WGS analysis until post hoc bioinformatic analysis according to the breakpoint from Nanopore platform.

- A deeper survey of the detail of the authors data should be given surrounding the breakpoint to make it explicit that its APCDD1 impacted. Does GJA1 change in expression? This gene had been implicated in eye disease phenotypes and the authors should be very explicit about whether GJA1 changes in expression and whether that is relevant to the phenotype present.

Our response: We thank the reviewer for this important comment. The expression of genes in the vicinity of the two breakpoints, including *FABP7*, *TBC1B32*, *VAPA*, *TXNDC2*, *RAB31*, *PIEZO2*, *NAPG*, *HSF2*, *SERINC1*, *PKIB* as well as *GJA1* were quantitated in iPSCs from the patient with CIA and are now shown in revised Supplementary Fig 8. *GJA1* showed a slight trend towards downregulation compared with that in the normal individual with a fold change of approximate 0.56, while others showed slight increases or decreases. While variants in *GJA1* are known to cause syndromes that include eye anomalies such as oculodentodigital dysplasia, to date these are either autosomal dominant missense mutations implying a gain of function or homozygous null mutations, which would not result from a 50% decrease in expression. In fact, a systematic review in our previous study suggests that heterozygous null alleles created by *GJA1* variants are well tolerated without a phenotype, suggesting that haploinsufficiency is not a pathogenic mechanism for *GJA1* variants⁷. The reduced expression of *GJA1* seen in this family patient is similar to that seen in patients with heterozygous null alleles and thus probably is not relevant to the phenotype. This result has been added in the Results and Discussion sections.

More generally, as the authors have gone to the excellent effort of generating Hi-C, RNA-seq and CUT&Tag in patient iPSCs, the authors should show:

*** Quantifications at individual replicate level of APCDD1 and all genes in the vicinity of the breakpoints (at least those in Fig 3). Was APCDD1 the only gene with dysregulated expression?**

Our response: We thank the reviewer for this excellent comment. *APCCD1* was the only gene with a statistically significant expression change as defined by an adjusted $P < 0.05$ and $|\log_2(\text{fold change})| > 1$. The quantifications at individual replicate level of all the 12 genes in the vicinity of the two breakpoints have been showed in the revised Supplementary Fig. 8 with the expression changes of 40 genes within 2.5 Mb of each breakpoint in the revised Supplementary Table 4. This information has also been added to the Results and Discussion sections as:

“By RNA-seq, the expression changes of genes in the vicinity of the two breakpoints were quantitated in iPSCs from the patient with CIA. The quantifications at individual replicate level of all the 12 genes close to the two breakpoints have been showed in the Supplementary Fig. 8 with the expression changes of all 40 genes within 2.5 Mb of each breakpoint in the Supplementary Table 4. The results showed that *APCCD1* was the only gene with a statistically significant expression change as defined by an

adjusted $P < 0.05$ and $|\log_2(\text{fold change})| > 1$, which we confirmed by real-time PCR and Simple Western analysis (Fig.3e-g).”

“*GJA1*, which is known to cause eye anomalies, was also located in the vicinity of the breakpoint. The transcription level of *GJA1* showed a slight trend towards downregulation in iPSCs from the patient with CIA compared with that in the normal individual with a fold change of approximate 0.56, which was not significant. While variants in *GJA1* are known to cause syndromes that include eye anomalies such as oculodentodigital dysplasia, to date these are either autosomal dominant missense mutations implying a gain of function or homozygous null mutations, which would not result from a 50% decrease in expression. In fact, a systematic review in our previous study suggests that heterozygous null alleles. created by *GJA1* variants are well tolerated without a phenotype, suggesting that haploinsufficiency is not a pathogenic mechanism for *GJA1* variants³⁰. The reduced expression of *GJA1* seen in this family patient is similar to that seen in patients with heterozygous null alleles and thus probably is not relevant to the phenotype.”

*** A zoom in of the Hi-C, RNA-seq, and CUT&Tag data surrounding their putative inducing enhancer and the breakpoints in patient and control iPSCs, realigning the data to the translocated sequence if necessary. The patient appears to have gained H3K27ac peaks, and the RNA-seq appears to show data spanning from APCDD1 to over the break point.**

Our response: The Hi-C, RNA-seq, and CUT&Tag data surrounding the breakpoints in Fig. 3 have been expanded in size so that details are legible as suggested.

*** The use of public data to evaluate the regulatory potential across different tissues to provide context is great, however, the main text should be made more explicit as to when public data is used. For example, when stated that CTCF and RAD21 were found at neo-TAD boundaries, it should be made clear that this is in public eye data and not patient iPSCs.**

Our response: The H3K27ac profiles across various tissues were obtained from the EpiMap project (accessed on May 15, 2023) and this has been added to the Results as:

“Furthermore, analysis of H3K27ac profiles across various tissues from the EpiMap project (accessed on May 15, 2023) revealed that the enhancer cluster on chromosome 6 exhibited tissue-specific activity,”

We have also explained that the binding profiles of CTCF and RAD21 were from the EpiMap database rather than the patient iPSCs, and this has been added to the Results section as:

" To investigate the binding profile of chromatin structural factors at the neo-TAD boundaries, we retrieved the binding profiles of two key factors, CCCTC-Binding Factor (CTCF) and Cohesin (indicated by its subunit RAD21), from the EpiMap database (accessed on May 15, 2023). "

No other binding profiles were taken from public databases.

*** The authors should supply the visualisation of public data at the chr18 locus too, are there eye specific enhancers implied to regulate APCDD1, what happens to these under the translocation.**

Our response: Thank you very much. Several clusters of enhancers in the vicinity of the breakpoint at chr18 were detected in iPSCs samples using H3K27ac CUT&Tag (Supplementary Fig.7). These enhancer clusters at chr18 showed ubiquitous presence in various tissues according to H3K27ac profiles from the EpiMap project (Supplementary Fig.7). These have been added in the Results section and revised Supplementary Fig.7 as suggested:

"Notably, in the TADs on chromosomes 6 and 18 of the unaffected individual, we detected clusters of enhancers using H3K27ac CUT&Tag. Furthermore, analysis of H3K27ac profiles across various tissues from the EpiMap project (accessed on May 15, 2023) revealed that the enhancer cluster on chromosome 6 exhibited tissue-specific activity (Supplementary Fig.6), whereas the enhancer clusters on chromosome 18 showed ubiquitous presence in various tissues (Supplementary Fig.7)."

- The authors should acknowledge the limitation that since they evaluate gene expression in pluripotent cells that the enhancer activity and gene expression changes may not precisely reflect those within the relevant cells within the iris.

Our response: We agree with the reviewer. Both gene expression and enhancer activity may be specific for individual tissues. The major limitation of this study is the unavailability of the patient iris tissue and thus the evaluation of gene expression and enhancer activity are necessarily based on iPSC samples. Although the iPSCs provide information for early development of human tissues, including eyes, a better disease model may be generating iris organoids based on iPSCs but this approach has yet to be developed. We have added this in the Discussion section:

"As both gene expression and enhancer activity may be specific for individual tissues, a major limitation of this study is the unavailability of iris tissue from the affected patients, which would allow evaluation of gene expression and enhancer activity so that these are necessarily based on iPSC samples. Although the iPSCs provide information for human tissues, including eyes, at early stages of development, a better disease model may be generating iris organoids based on iPSCs. However, this approach is not yet technically feasible."

- The discussion of the literature and role of APCDD1 is limited, and whilst it relates to the retina it feels necessary that the authors cite and consider:

Mazzoni J, Smith JR, Shahriar S, Cutforth T, Ceja B, Agalliu D. The Wnt Inhibitor *Apcdd1* Coordinates Vascular Remodeling and Barrier Maturation of Retinal Blood Vessels. *Neuron*. 2017 Dec 6;96(5):1055-1069.e6. doi: 10.1016/j.neuron.2017.10.025. Epub 2017 Nov 16. PMID: 29154126; PMCID: PMC5728434.

Our response: The discussion of literature and role of *APCDD1* has been expanded and revised as follows:

“*APCDD1* was initially isolated in 2002 as a target of the beta-catenin/T-cell factor 4 complex³⁸. Subsequent studies found that it functions as an inhibitor of Wnt signalling and further as a dual inhibitor of both Wnt and BMP signalling in the developing nervous system and skin^{14,39}. Moreover, *Apcdd1* has been reported to coordinate vascular development of retinal blood vessels by modulating Wnt/Norrin signalling activity in the mouse⁴⁰. These results suggest that *APCDD1* has complicated functions central to animal development and human disease. It is important to note that the phenotype in this family appears to result from upregulation of *APCDD1* in patients with CIA, which is contrary to the dominant negative pathogenic mechanism of the Leu9Arg variant seen in previous studies.^{14,39} These results suggest that *APCDD1* has a complex function, both in the iris and in the hair follicle, and must be precisely regulated for proper development.”

This should be done considering the authors work and Shinomura et al, 2010. The section on WLS is speculative and unless the authors wish to provide further results or supplemental material to support this, the comment should be made in the discussion rather than appear as a result.

Our response: We agree with the reviewer. This comment has been revised and moved to the Discussion with a statement of the need for further studies:

“Moreover, it has previously been shown that *APCDD1* can interact in vitro with WNT3A and LRP5, both of which are essential components of Wnt signalling.¹⁴ However, neither LRP5 nor WNT3A were detected in the human iris in a previous study on cell atlas generation of the human ocular anterior segment. At least 625 proteins have been implicated in the Wnt signalling pathway (Uniprot, <https://www.uniprot.org/>), of which 136 coding genes are expressed in fibroblasts of the human iris, including *APCDD1*. By RNA-seq analysis five of the 136 genes show a significantly altered expression in iPSCs in the patient as compared to the control, including *APCDD1*, *GREM1*, *RSPO2*, *WIF1*, and *WLS*. Moreover, it is noted that variants in *WLS*, encoding *wntless*, have been reported to be associated with iris coloboma¹⁶. This suggests that *APCDD1* and *WLS* might possibly interact or share a similar pathway, although this needs to be validated in further studies.”

- Can the authors be more explicit on the significance of the additional families with CIC and how these relate to the implied up-regulation of APCDD1. Are these all relevant to be included in this study?

Our response: Although additional four families with CIA were identified in this study, the genetic basis was only identified in one proband, namely $inv(18)(p11.2q11.2)$. It is noted that the inversion also occurred at $chr18p11.2$, the same location as the translocation seen in the large family. Therefore, the potential mechanism of the CIA in the patient with inversion variation may also be related to upregulation of *APCDD1*. The pathogenic mechanisms in the remaining three families have not been delineated and identification of the precise genetic mechanisms responsible for the disease in these families will be a subject for future studies. This has been added in the Discussion section:

“In addition to the original large family, four additional families with CIA were identified in this study, although the genetic basis was only identified in one proband, namely a de novo balanced inversion $inv(18)(p11.2q11.2)$. It is noteworthy that the inversion also occurred at $chr18p11.2$, the same location as the translocation seen in the original family. Therefore, the potential mechanism of the CIA in the patient with inversion variation may also be related to upregulation of *APCDD1*. However, the pathogenic mechanisms in the remaining three families have not been delineated and identification of the precise genetic mechanisms responsible for the disease in these families will be a subject for future studies.”

- The discussion should perhaps underscore more clearly that as APCDD1 is already expressed in the tissue of interested and expression is hypothesised to increase in patient with the translocation that CIC is not expected to result from loss-of-function of APCDD1.

Our response: We apologize for being unclear. This has now been explained in the Discussion as follows:

“It is important to note that the phenotype in this family appears to result from upregulation of *APCDD1* in patients with CIA, which is contrary to the dominant negative pathogenic mechanism of the Leu9Arg variant seen in previous studies^{14,39}.”

Reviewer #3 (Remarks to the Author):

This well-written manuscript reports a new ocular phenotype attributable to a balanced reciprocal chromosomal translocation. Key strengths include the authors convincing demonstration that the translocation generates new

chromatin topologically associated domains, juxtaposing enhancer elements adjacent to the candidate causative gene. Fully validated, this finding would demonstrate a structural chromosomal variant in a non-exonic region causes a novel Mendelian disease. However, several areas require enhancement, with one additional experiment, to fully support the conclusions. I hope the following assist the authors with their revisions:

1) The phenotype reported is not of a coloboma; a group of disorders caused by failure of closure of the choroid fissure (please google 'coloboma image' and compare with Fig. 1b). Rather than implicating the ~35th gene in coloboma, the authors have in fact identified a more interesting phenotype, where most of the iris appears not to develop.

Our response: We agree. The phenotype has been revised as central iris aplasia (CIA).

Apart from correcting the text, the authors need to illustrate the ocular anterior segment in greater detail, and at higher resolution for most/all the affected individuals. This would reveal any phenotypic variability. They could beneficially include multiple additional data [anterior chamber depth, central corneal thickness, endothelial cell density, axial length measurements, etc.]. The iris induces corneal development, so some of these parameters, and potentially the trabecular meshwork and or intraocular pressure may be altered. If UBM imaging is feasible, the manuscript would provide a very detailed first report of a novel phenotype.

Our response: We thank the reviewer for this suggestion. Unfortunately, detailed clinical information of the ocular anterior segment was available from only three patients in this family, whereas UBM imaging and corneal endothelial cell density were obtained from only a single patient. This family was ascertained 10 years previously and all but one are resistant to returning for further studies. The requested data have been added to the Results section with the Supplementary Table 1 and The Supplementary Fig. 1 as follows:

“All six affected members had clear cornea and normal-like fundus with foveal reflex (Fig. 1b-e and Supplementary Fig. 1c-h) as normal controls (Fig.1 f-i). The axial length of the six affected individuals ranged from 19.17 mm to 25.32 mm. The corneal horizontal diameters, central cornea thickness, intraocular pressure, and central anterior chamber depth were available from three patients (IV:6, V:6, and V:4) and all of these parameters are in the normal range (Supplementary Table 1). Ultrasound biomicroscopy was performed on one patient (V:6), in which the anterior chamber angle is open (Fig. 1c and Supplementary Fig. 1i,j). Gonioscopy examination of this patient suggested a mild malformation of the anterior chamber angle, that is an increasing presence of pectinate ligament (Supplementary Fig. 1k,l). The corneal endothelial cell density of the patient V:6 was 3003 cells/mm² with hexagonal cells of 68% in the right eye and 2892 cells/mm² with hexagonal cells of 73% in the left eye (Supplementary Fig. 1m,n). Two of the six, III:3 and IV:4, had

age-related cataract on examination at ages of 72 and 55 years old, respectively (Supplementary Table 1).”

Supplementary Table 1. Clinical data of affected members from the family.

ID	Age at	Gender	BCVA	Pupil size (mm)	IOP (mmHg) OD;OS	Corneal parameter (mm)		ACD OD;OS (mm)	Lens	Axial length OD;OS (mm)
	Exam (yrs)					Diameter OD;OS	CCT OD;OS			
III:3	72	F	0.1*	8	/	/	/	/	Age-related cataract	24.77;24.13
IV:4	55	F	0.3*	8	/	/	/	/	Age-related cataract	25.32;25.31
IV:6	39	F	0.5	8	16.0;14.9	12; 12	0.61;0.61	2.76;2.80	/	21.21;21.55
V:1	37	M	0.6	8	/	/	/	/	/	23.11;23.14
V:4	31	M	0.6	6	17;15	11; 11	0.58;0.60	2.63;2.67	/	23.00;22.87
V:6	6	F	0.6	8	14.0;16.0	12; 12	0.55;0.55	2.55;2.41	/	19.35;19.17

Notes: BCVA, best corrected visual acuity. OD, right eye. OS, left eye. IOP, intraocular pressure. CCT, central corneal thickness.

ACD, anterior chamber depth. *, the lower BCVAs in the two patients are due to partially lens opacity of age-related cataract.

All the patients had the first symptom of photophobia with onset during first few months after birth.

2) The fluorescent in situ data, and most important part of Figure 2, is small and tough to read. Larger and enhanced resolution panels highlighting the normal autosome and both structurally variant chromosomes, would assist. These three should also appear as larger insets, ideally with similar color labelling to Fig 2a, to demonstrate each rearrangement. The authors could consider FISH with 3 or 4 differentially-labelled probes to both chromosome 6 and 18, to illustrate the rearrangements' complexity [currently separate images are shown for chr.6 and then chr. 18].

Our response: The FISH was performed using three-color probes, namely aqua probes labelling the centromeres of chr6 and chr18, a red probe labelling the chr6qter, and a green probe labelling the chr18pter. Larger panels in the figure highlighting the normal and structurally variant chromosomes are provided as follows:

In these figures chromosomes 6 and 18 are distinguishable by their centromeric fluorescent labels, their telomeric fluorescent labels and their structure, chromosome 6 being a member of group C (medium sized submetacentric) and chromosome 18 being a member of group E (short metacentric). In the upper panel, aqua signals labelled centromere of chromosome 6, red fluorescent signals indicate 6qter at normal chromosome 6 and derivative chromosome 18, whereas green fluorescent signals indicate 18pter at normal chromosome 18 and derivative chromosome 6. In the lower panel, aqua signals labelled centromere of the chromosome 18, red fluorescent signals indicate 6qter at the normal chromosome 6 and derivative chromosome 18, whereas green fluorescent signals indicate 18pter at the normal chromosome 18 and derivative chromosome 6. A diagram showing normal and structurally variant chromosomes labelled by probes in each panel was provided on the right for clarity.

3) Analysis of the structural variant with high-throughput chromosome confirmation capture, and then CUT & Tag sequencing, was viewed as very

strong. However, there appears a logic leap to APCDD1 being the causative gene/sole causative gene, based on its position and upregulated expression on RNA-sequencing and quantitative immunoblotting. The current volcano plot while nice, is not particularly relevant if over-expression of a wnt-bmp inhibitor dysregulates numerous downstream genes. Instead, we need to see whether other genes adjacent to the breakpoints are dysregulated, or if APCDD1 is the only candidate. This would be addressed by providing the full RNA-seq dataset [as a supplemental table], and a list of genes within 2.5Mb of each breakpoint with their degree of altered expression.

Our response: We thank the reviewers for excellent comment. The full data set of differential expression from RNA-seq has been provided as Supplementary Data as suggested. As we also addressed the response to the third comment of Reviewer #2, changes in the expression of genes in the vicinity of the two breakpoints were quantitated in iPSCs from the patient with CIA. The expression changes of all 40 genes within 2.5 Mb of each breakpoint are listed in the Supplementary Table 4. The results show that *APCCD1* is the only gene with a statistically significant expression change as defined by an adjusted $P < 0.05$ and a $|\log_2(\text{fold change})| > 1$, which we confirmed by real-time PCR and Simple Western analysis (Fig. 3e-g).

Once the above has been tackled, a key experiment is missing. The authors should demonstrate whether CRISPR interference targeting the enhancer elements identified via CUT & Tag, abrogates increased APCDD1 mRNA expression. Evidence of partial normalization of APCDD1 expression with CRISPRi / multiplexed CRISPRi would neatly validate the central mechanism proposed in this paper

Our response: We thank the reviewers for the suggestion. As we addressed the comment from the Reviewer #1. Since it is difficult to knockout the whole 200 kb enhancer cluster, an 8.8 kb region around the first H3K27ac peak was targeted to knock out this region in iPSCs based on the cells from the patient with translocation (Fig. 4a). This is actually a doubly heterozygous knockout of 10.6 kb with the endpoints varying by 1 bp (Fig. 4b,c). RT-qPCR results show that overexpression of *APCDD1* RNA was partially rescued in iPSCs with enhancer knockout comparing with that in iPSCs with the translocation but without enhancer knockout (Fig. 4d). This result supports the potential mechanism that the ectopic chromatin interactions between *APCDD1* and enhancers results in the upregulation of *APCDD1*. These results have been added to the Results and Methods sections as well as the revised Fig. 4.

4) The zebrafish data are not compelling, and several concerns exist.

First, as frequently observed, there are two zebrafish paralogs of the mammalian ortholog. However, this complexity is neither discussed, nor is the second ocular-expressed paralog investigated (only *apcdd1*-like).

Our response: The reviewer is correct concerning the complexity of the zebrafish system. We have now investigated both paralogues of *APCDD1* and found that they have overlapping functions as follows. The mRNA of *LOC110438155*, the second paralog of *APCDD1* in zebrafish, was synthesized *in vitro* and microinjected into zebrafish eggs. At 3dpf, a delayed fusion of the choroid fissure was also observed in 55.1% of the larvae with overexpression of the *LOC110438155*, which is similar to the results seen with *apcdd1*-like, and significantly higher than that in wild-type and standard control larvae. This suggests that the two *APCDD1* paralogues have overlapping functions similar to *APCDD1* in the human and confirm the role of *APCDD1* in development of the iris. These results have been added in the Results and revised Supplementary Fig.11:

“To validate the association between iris phenotypes and upregulation of *APCDD1*, mRNA of two zebrafish paralogues of human *APCDD1*, *apcdd1l* and *LOC110438155*, were synthesised and injected into zebrafish eggs at the one-cell stage. A phenotype of ocular coloboma was observed in 41.6% of *apcdd1l* overexpressed larvae and 55.1% of *LOC110438155* overexpressed larvae at day 3 post-fertilization (3dpf), which were significantly higher than that in wild-type (wt) larvae as well as in larvae injected with standard control mRNA (std) (Supplementary Fig. 11). By immunostaining with anti-phalloidin, a clear separation of the basement membrane at the optic fissure region was marked in the two groups of larvae with *apcdd1* overexpression, whereas the basement membrane was continuous in wt and std larvae at 3dpf (Supplementary Fig. 11). The coloboma due to failure of optic fissure closure observed in *apcdd1* overexpressed larvae confirms the role of *APCDD1* in development of the iris.”

Second, the logic of injecting mRNA of a dual bmp and wnt inhibitor into a one cell zebrafish embryo to model tissue and temporally-specific human APCDD1 over-expression, is unclear. The inhibitor’s broad roles cause profound effects at picogram doses, that do not recapitulate the human phenotype [compare Figs. 1b and 4c].

Our response: As *APCDD1* is essential for early development of zebrafish as a dual bmp and wnt inhibitor, it is expected to cause widespread profound effects, including in the eyes, as described in a previous studies.⁸ However, at the levels used in this study the effects of *apcdd1* appear to be relatively specific for eye development, as seen in the revised Supplementary Fig. 11. While this does not exactly recapitulate the human phenotype seen in Fig. 1b, taken with the other data in the manuscript, the zebrafish results strongly support a causative role for *APCDD1* in iris closure, irrespective of additional extraocular effects.

Third, the observed delayed fusion of the zebrafish choroid fissure [a coloboma], reflects altered patterning and not a tissue-specific defect. BMP and Wnt signaling have essential roles establishing and maintaining the dorso-ventral axis of the retina/eye, so an inhibitor will alter levels of multiple

bmp/wnt ligands, inducing colobomata without supporting the human phenotype.

Achieving temporal and spatially-restricted mRNA over-expression in the zebrafish iris would be challenging. Even if feasible, it may not recapitulate the human phenotype due to multiple paralogs, differences in developmental signaling pathways, and 450 million years of evolutionary separation. I will be interested in the other reviewers' comments on this point. Since I think it would be unreasonable to require in vivo electroporation in murine/xenopus, a better approach may be to discuss the challenges modelling a disorder caused by highly specific dysregulation of a single gene during a restricted temporal interval, with mRNA micro-injection of one cell zebrafish embryos. Fig 4 could then be relegated to supplemental, to demonstrate that this has been attempted, but proved unsuccessful.

Our response: We appreciate and understand the reviewer's concerns. We have thus moved all the zebrafish data into supplemental information and added a detailed discussion of the conceptual and practical difficulties in using the zebrafish as a model system for this family to the Discussion:

"In addition, it is challenging to recapitulate the CIA phenotype in a zebrafish model with overexpression of *APCDD1* paralogs because of the difficulty of achievement of temporal and spatially restricted *apcdd1* overexpression in the zebrafish iris. In this study, mRNAs of *APCDD1* paralogs were microinjected in one-cell eggs of zebrafish to generate models with overexpression of the gene. However, the coloboma phenotype in larvae with *apcdd1* overexpression is different from the human phenotype although it suggests a role in development of the iris. Even if feasible, it may not recapitulate the human phenotype due to multiple paralogs, differences in developmental signalling pathways, and 450 million years of evolutionary separation."

References

- 1 Misteli, T. The Self-Organizing Genome: Principles of Genome Architecture and Function. *Cell* **183**, 28-45, doi:10.1016/j.cell.2020.09.014 (2020).
- 2 Ghavi-Helm, Y. *et al.* Highly rearranged chromosomes reveal uncoupling between genome topology and gene expression. *Nature Genetics* **51**, 1272-+, doi:10.1038/s41588-019-0462-3 (2019).
- 3 Ho, S. V. S., Urban, A. E. & Mills, R. E. Structural variation in the sequencing era. *Nature Reviews Genetics* **21**, 171-189, doi:10.1038/s41576-019-0180-9 (2020).
- 4 Huddleston, J. *et al.* Discovery and genotyping of structural variation from long-read haploid genome sequence data. *Genome Research* **27**, 677-685, doi:10.1101/gr.214007.116 (2017).
- 5 Alkan, C., Coe, B. P. & Eichler, E. E. Genome structural variation discovery and genotyping. *Nature Reviews Genetics* **12**, 363-376, doi:10.1038/nrg2958 (2011).
- 6 Bouwman, A. C., Derks, M. F. L., Broekhuijse, M. L. W. J., Harlizius, B. & Veerkamp, R. F. Using short read sequencing to characterise balanced reciprocal translocations in pigs. *Bmc Genomics* **21**, doi:10.1186/s12864-020-06989-x (2020).

- 7 Li, X. Q. *et al.* Heterozygous GJA1 variants with ocular phenotype: Missense in domain but truncation out of domain. *Molecular Vision* **27**, 309-322 (2021).
- 8 Vonica, A. *et al.* Apcdd1 is a dual BMP/Wnt inhibitor in the developing nervous system and skin. *Dev Biol* **464**, 71-87, doi:10.1016/j.ydbio.2020.03.015 (2020).

REVIEWER COMMENTS

Reviewer #1 (Remarks to the Author):

The authors have addressed all my concerns and provide beautiful additional enhancer KO data. I have no further questions and would like to congratulate the authors on a really nice paper.

Reviewer #2 (Remarks to the Author):

Overall, the reviewers have addressed many of my comments. However, I think the conclusion that APCCD1 is the single gene implicated by their assays is not well supported by the authors' data. There are issues in the interpretation of the RNA-seq data, particularly, the interpretation and language surrounding GJA1 claiming it has a "slight trend towards downregulation" is extremely odd and is not supported by the data presented. These statements need to be altered before the paper can be accepted.

GJA1 appears to be significantly downregulated. GJA1 has the smallest FDR in Supp. Table 4, it moreover has 12th smallest FDR in supplementary dataset 1. Notably, there is a discrepancy between these tables as the FDR and fold changes are different for GJA1 log2 FC and FDR: Supp. Table 4: -0.809, 1.58e-93; Supp. Dataset 1: -0.8143, 1.43e-103. Please can this be corrected.

GJA1 shows a 0.56x reduction and is excluded from the lists of differentially expressed genes by a $|\log_2\text{Foldchange}| > 1$ threshold ($|\text{FC}| > 2$). The use of fold change thresholds (effect size) in bulk RNA-seq is subjective and therefore not universally agreed. There are multiple issues with this: first, $|\text{FC}| > 2$ is unusually high; second, fold change is a statistic that favours the lowly expressed APCCD1 and penalises the very highly expressed GJA1, achieving a $|\text{FC}| > 2$ is much harder for GJA1. Third, filtering on fold change is generally an acceptable procedure to define a set of strongly differentially expressed genes for gene set enrichment, but it not appropriate to determine precisely which genes are differentially expressed. This is problematic because we do not know what change in expression is relevant to the cell. Moreover, this is a bulk assay reporting average fold change over the population, there may be subpopulations of cells with strong reduction of GJA1. I am not suggesting the authors employ single-cell methods, rather that they relax their fold change threshold and consider the interpretations of all genes differentially expressed.

What seems to be the case, and on the face of it seems a very nice result, is that the translocation shifts an enhancer from GJA1 leading to a reduction in expression ($\text{FC} = 0.56$, $\text{FDR} = 5.71\text{e-}6$), to APCDD1 leading to an increase in expression ($\text{FC} = 1.403$, $\text{FDR} = 5.6\text{e-}27$). This translocation and refolding of chromatin may lead to other gene expression changes such as the increase in PIEZO2 and NAPG and the decrease in FABP7.

The authors should reduce their fold change threshold, report that plainly how each gene's expression is altered for all genes in the vicinity of the translocation. This only need be a change in the text. Then the authors should provide rationale for why APCDD1 is the most likely to cause disease. This will necessitate a more in depth discussion of GJA1 in the least. It is good that the authors highlight that the known eye anomalies caused by GJA1 defects would not result from a 50% reduction in expression, but their statement that "autosomal dominant missense mutations implying a gain of function" is not correct and needs clarification.

Given that the authors' main interpretation of APCDD1 is in disruption of WNT signalling, it seems relevant that GJA1 (Cx43) interacts with the WNT pathway and it physically interacts with beta-catenin showing and may be a co-transcriptional activator (<https://pubmed.ncbi.nlm.nih.gov/31067079/>). The authors' should engage in the possibility that the disease could be caused by the reduction in GJA1 expression alone or concomitant with an increase in APCDD1. Perhaps the most compelling way for

the authors to assert the importance of APCDD1 over GJA1 would be to stain for GJA1 and show that its expression is unlikely to be relevant to the phenotype. I would allow the editors and other reviewers to determine whether this was necessary.

Reviewer #3 (Remarks to the Author):

The authors have been receptive to the reviews, beneficially revising their manuscript. In terms of the major points that I raised – the FISH data are substantially improved and are accessible to non-specialist readers, and inclusion of data showing the effect of deleting the enhancer supports the key hypothesis.

While comfortable with the manuscript moving forward, I encourage the authors to consider small textual edits in three areas, especially #3:

1) Partial rescue vs rescue:

At some points they state that overexpression of APCDD1 RNA was partially rescued, while at others they state “rescue”. Since Fig. 4D demonstrates partial rescue, this seems the more appropriate term.

2) Potential role for GJA1 (and other genes)

Inclusion of new data (Supp Table 4) demonstrate dysregulation of other genes, including GJA1 which has roles in iris/anterior segment development. The authors made quite extensive textual changes to the Discussion (that could be abridged). It may be beneficial to also add a line stating, that while their data support a model where dysregulation of APCDD1 is primarily responsible for the phenotype, contributions from other adjacent genes, such as GJA1, cannot be excluded.

3) Aplasia?

The authors revised description of the iris phenotype, which is good.

However, they should triple check whether ‘aplasia’ is the optimal term – it suggests complete absence of tissue, whereas the phenotype appears less severe (Supp. Fig 1a/b shows an enlarged pupil with loss of some iris tissue, but preservation of the the iris ruff). Only the authors have examined these patients, who appear to have some variability in phenotypic severity, so other terms may be more appropriate (perhaps hypoplasia?).

The terminology they select will be used for centuries, so I am sure they will make their choice carefully and wisely (avoiding the abbreviation ‘CIA’ may be a bonus).

Thank you for the opportunity of reviewing this manuscript.

Authors' Response to Comments from the Editors and the Reviewers

In the following we list the comments from the editor and reviewers in **bold** followed by our changes/explanations in plain text. Changes made in the manuscript are highlighted in blue.

REVIEWER COMMENTS

Reviewer #1 (Remarks to the Author):

The authors have addressed all my concerns and provide beautiful additional enhancer KO data. I have no further questions and would like to congratulate the authors on a really nice paper.

Our response: Thanks a lot for your help and confirmation.

Reviewer #2 (Remarks to the Author):

Overall, the reviewers have addressed many of my comments. However, I think the conclusion that APCCD1 is the single gene implicated by their assays is not well supported by the authors' data. There are issues in the interpretation of the RNA-seq data, particularly, the interpretation and language surrounding GJA1 claiming it has a "slight trend towards downregulation" is extremely odd and is not supported by the data presented. These statements need to be altered before the paper can be accepted.

GJA1 appears to be significantly downregulated. GJA1 has the smallest FDR in Supp. Table 4, it moreover has 12th smallest FDR in supplementary dataset 1. Notably, there is a discrepancy between these tables as the FDR and fold changes are different for GJA1 log₂ FC and FDR: Supp. Table 4: -0.809, 1.58e-93; Supp. Dataset 1: -0.8143, 1.43e-103. Please can this be corrected.

GJA1 shows a 0.56x reduction and is excluded from the lists of differentially expressed genes by a $|\log_2\text{Foldchange}| > 1$ threshold ($|\text{FC}| > 2$). The use of fold change thresholds (effect size) in bulk RNA-seq is subjective and

therefore not universally agreed. There are multiple issues with this: first, $|FC| > 2$ is unusually high; second, fold change is a statistic that favours the lowly expressed *APCCD1* and penalises the very highly expressed *GJA1*, achieving a $|FC| > 2$ is much harder for *GJA1*. Third, filtering on fold change is generally an acceptable procedure to define a set of strongly differentially expressed genes for gene set enrichment, but it not appropriate to determine precisely which genes are differentially expressed. This is problematic because we do not know what change in expression is relevant to the cell. Moreover, this is a bulk assay reporting average fold change over the population, there may be subpopulations of cells with strong reduction of *GJA1*. I am not suggesting the authors employ single-cell methods, rather that they relax their fold change threshold and consider the interpretations of all genes differentially expressed.

What seems to be the case, and on the face of it seems a very nice result, is that the translocation shifts an enhancer from *GJA1* leading to a reduction in expression ($FC = 0.56$, $FDR = 5.71e-6$), to *APCDD1* leading to an increase in expression ($FC = 1.403$, $FDR = 5.6e-27$). This translocation and refolding of chromatin may lead to other gene expression changes such as the increase in *PIEZO2* and *NAPG* and the decrease in *FABP7*.

The authors should reduce their fold change threshold, report that plainly how each gene's expression is altered for all genes in the vicinity of the translocation. This only need be a change in the text. Then the authors should provide rationale for why *APCDD1* is the most likely to cause disease. This will necessitate a more in depth discussion of *GJA1* in the least. It is good that the authors highlight that the known eye anomalies caused by *GJA1* defects would not result from a 50% reduction in expression, but their statement that "autosomal dominant missense mutations implying a gain of function" is not correct and needs clarification.

Given that the authors' main interpretation of *APCDD1* is in disruption of WNT signalling, it seems relevant that *GJA1* (*Cx43*) interacts with the WNT pathway and it physically interacts with beta-catenin showing and may be a co-transcriptional activator (<https://pubmed.ncbi.nlm.nih.gov/31067079/>). The authors' should engage in the possibility that the disease could be caused by the reduction in *GJA1* expression alone or concomitant with an increase in *APCDD1*. Perhaps the most compelling way for the authors to assert the importance of *APCDD1* over *GJA1* would be to stain for *GJA1* and show that its expression is unlikely to be relevant to the phenotype. I would allow the editors and other reviewers to determine whether this was necessary.

Our response: We completely understand your concern about the contribution of *GJA1* downregulation. We agree with you that the use of fold change thresholds in RNA-seq analysis is arbitrary and may penalise highly expressed genes, such as *GJA1* in this study. In fact, we indeed first focused on *GJA1* as soon as we analyzed the RNA-seq data because it is well known that variants in *GJA1* are associated with

oculodental dysplasia involving iris anomaly. The RNA-seq data suggested a 0.44x (FC=0.56) reduction of *GJA1* RNA in iPSCs of the patient with a significant FDR. The approximately half reduction of *GJA1* RNA might be expected to have an effect similar to that of a heterozygous null allele of the gene, which is well tolerated in human beings since individuals carrying heterozygous null allele are unaffected according to a systematic review in our previous study¹. Therefore, *GJA1* was excluded from being primarily responsible for the phenotype considering both the mechanism and the transcriptional level of the gene rather than merely a rough analysis with a threshold of $|FC| > 2$.

When the RNA-seq data were further analyzed to explore the gene responsible for the phenotype, a threshold of an adjusted $P < 0.05$ and a $|\log_2(\text{fold change})| > 1$ was used because it is common in most of previous studies, whereas some studies set a higher threshold, such as $|\log_2(\text{fold change})| > 1.5$ or even > 2 . We also worried that an unusual but lower criterion might tend to confuse the readers. However, we agree with your concern that expression of additional genes with lower fold changes might also contribute to the phenotype considering the complex translocation. To account for this, we have modified the Results section to explain additional changes in the expression of adjacent genes as follows:

“When the threshold was relaxed as $|\log_2(\text{fold change})| > 0.5$ (FC > 1.4 or FC < 0.7), six additional genes, including *TBC1D32*, *GJA1*, *FABP7*, *TRDN*, *NDUFV2*, and *PIEZO2*, show a significant changes in expression, especially a 0.44x reduction (FC = 0.56) of *GJA1* RNA, which has the smallest FDR of 1.58E-93 (Supplementary Table 4).”.

In addition, change in *APCDD1* is likely to be the primary mechanism for the phenotype according to series analyses, including neo-TADs from Hi-C analysis, ectopic interaction of the enhancer cluster with *APCDD1* by CUT&Tag analysis, significant expression changes by RNA-seq, and the specific staining in the iris. However, we agree with you that a possible contribution of decreased *GJA1* expression cannot be excluded and have modified the Discussion as follows:

“There currently is no strong evidence suggesting that a 50% reduction of *GJA1* or the other adjacent genes is associated with a human phenotype. However, while our data suggest that upregulation of *APCDD1* is likely a primary mechanism responsible for the phenotype, possible contributions from expression changes in adjacent genes, either from a single gene (e.g. *GJA1*) or the combined effects of several genes cannot be excluded. In this regard, it is interesting to note that WNT signalling causes the *GJA1* product, Cx43, to translocate to the nucleus similar to its action on β -catenin, with which Cx43 interacts directly, suggesting that it might participate in the β -catenin destruction complex or perhaps serve as a co-transcriptional activator².”

We also appreciate your comments about several statements and have revised our manuscript accordingly, including:

- 1) “slight trend towards downregulation” have been revised as: “downregulation to approximately half”.
- 2) The supplementary dataset has been replaced by a correct version. The similar with mildly different results between the two versions are from of the different cutoffs for excluding the impact of low or non-expressed genes. The correct version was generated using the method as described in the Methods section, in which genes were removed when their cumulative reads count from all replicates were less than 20.
- 3) “autosomal dominant missense mutations implying a gain of function” has been revised as “heterozygous missense variants implying a gain of deleterious function or a dominant-negative effect”

Reviewer #3 (Remarks to the Author):

The authors have been receptive to the reviews, beneficially revising their manuscript. In terms of the major points that I raised – the FISH data are substantially improved and are accessible to non-specialist readers, and inclusion of data showing the effect of deleting the enhancer supports the key hypothesis.

While comfortable with the manuscript moving forward, I encourage the authors to consider small textual edits in three areas, especially #3:

1) Partial rescue vs rescue:

At some points they state that overexpression of APCDD1 RNA was partially rescued, while at others they state “rescue”. Since Fig. 4D demonstrates partial rescue, this seems the more appropriate term.

Our response: Thank you very much for pointing out this. This has been corrected accordingly.

2) Potential role for GJA1 (and other genes)

Inclusion of new data (Supp Table 4) demonstrate dysregulation of other genes, including GJA1 which has roles in iris/anterior segment development. The authors made quite extensive textual changes to the Discussion (that could be abridged). It may be beneficial to also add a line stating, that while their data support a model where dysregulation of APCDD1 is primarily responsible for the phenotype, contributions from other adjacent genes, such as GJA1, cannot be excluded.

Our response: Thank you for your kind suggestion. This has been revised accordingly. Please see our response to Reviewer 2 as follows:

We completely understand your concern about the contribution of *GJA1* downregulation. We agree with you that the use of fold change thresholds in RNA-seq analysis is arbitrary and may penalise highly expressed genes, such as *GJA1* in this study. In fact, we indeed first focused on *GJA1* as soon as we analyzed the RNA-seq data because it is well known that variants in *GJA1* are associated with oculodental dysplasia involving iris anomaly. The RNA-seq data suggested a 0.44x (FC=0.56) reduction of *GJA1* RNA in iPSCs of the patient with a significant FDR. The approximately half reduction of *GJA1* RNA might be expected to have an effect similar to that of a heterozygous null allele of the gene, which is well tolerated in human beings since individuals carrying heterozygous null allele are unaffected according to a systematic review in our previous study¹. Therefore, *GJA1* was excluded from being primarily responsible for the phenotype considering both the mechanism and the transcriptional level of the gene rather than merely a rough analysis with a threshold of $|\text{FC}| > 2$.

When the RNA-seq data were further analyzed to explore the gene responsible for the phenotype, a threshold of an adjusted $P < 0.05$ and a $|\log_2(\text{fold change})| > 1$ was used because it is common in most of previous studies, whereas some studies set a higher threshold, such as $|\log_2(\text{fold change})| > 1.5$ or even > 2 . We also worried that an unusual but lower criterion might tend to confuse the readers. However, we agree with your concern that expression of additional genes with lower fold changes might also contribute to the phenotype considering the complex translocation. To account for this, we have modified the Results section to explain additional changes in the expression of adjacent genes as follows:

“When the threshold was relaxed as $|\log_2(\text{fold change})| > 0.5$ (FC > 1.4 or FC < 0.7), six additional genes, including *TBC1D32*, *GJA1*, *FABP7*, *TRDN*, *NDUFV2*, and *PIEZO2*, show a significant changes in expression, especially a 0.44x reduction (FC = 0.56) of *GJA1* RNA, which has the smallest FDR of 1.58E-93 (Supplementary Table 4).”

In addition, change in *APCDD1* is likely to be the primary mechanism for the phenotype according to series analyses, including neo-TADs from Hi-C analysis, ectopic interaction of the enhancer cluster with *APCDD1* by CUT&Tag analysis, significant expression changes by RNA-seq, and the specific staining in the iris. However, we agree with you that a possible contribution of decreased *GJA1* expression cannot be excluded and have modified the Discussion as follows:

“There currently is no strong evidence suggesting that a 50% reduction of *GJA1* or the other adjacent genes is associated with a human phenotype. However, while our data suggest that upregulation of *APCDD1* is likely a primary mechanism responsible for the phenotype, possible contributions from expression changes in adjacent genes, either from a single gene (e.g. *GJA1*) or the combined effects of several genes cannot be excluded. In this regard, it is interesting to note that WNT signalling causes the *GJA1* product, Cx43, to translocate to the nucleus similar to its action on β -catenin, with which Cx43 interacts directly, suggesting that it might participate in the β -catenin destruction complex or perhaps serve as a co-transcriptional activator².”

3) Aplasia?

The authors revised description of the iris phenotype, which is good.

However, they should triple check whether ‘aplasia’ is the optimal term – it suggests complete absence of tissue, whereas the phenotype appears less severe (Supp. Fig 1a/b shows an enlarged pupil with loss of some iris tissue, but preservation of the the iris ruff). Only the authors have examined these patients, who appear to have some variability in phenotypic severity, so other terms may be more appropriate (perhaps hypoplasia?).

The terminology they select will be used for centuries, so I am sure they will make their choice carefully and wisely (avoiding the abbreviation ‘CIA’ may be a bonus).

Our response: We completely agree with you. The phenotype has been revised as “central iris hypoplasia” throughout the whole manuscript.

Thank you for the opportunity of reviewing this manuscript.

Our response: We appreciate your kind and valuable suggestions for improving our paper.

References

- 1 Li, X. Q. *et al.* Heterozygous GJA1 variants with ocular phenotype: Missense in domain but truncation out of domain. *Molecular Vision* **27**, 309-322 (2021).
- 2 Hou, X. *et al.* Wnt signaling regulates cytosolic translocation of connexin 43. *American journal of physiology. Regulatory, integrative and comparative physiology* **317**, R248-r261, doi:10.1152/ajpregu.00268.2018 (2019).

REVIEWERS' COMMENTS

Reviewer #2 (Remarks to the Author):

The amendments have largely addressed my concerns. I would prefer the authors to simply present that GJA1 is down and not present it as two thresholds, particularly given that this is in iPS cells and the fold changes may differ in the disease relevant cell. However, the authors' presentation is sufficient and I would be happy to see this published, although three minor points are below:

Three minor comments:

- Why is the greyed zone in Fig 3c at +/- log₂ 0.8? This coincides with GJA1, but the paper and the legend states that its +/- log₂ 1? This looks like the threshold has been lowered to the FC of GJA1.
- In general a heterozygous null does not imply a 50% reduction in expression. However, the paper contains a stronger statement than the reviewer's response. "However, a heterozygous null allele with a 50% decrease in expression is well tolerated in human beings since individuals carrying heterozygous null allele are unaffected according to a systematic review in our previous study²⁷" I do believe that ref. 27 assesses expression and so that authors should highlight that the 50% decrease may accompany the heterozygous null.
- GJA1 has a log₂ FC of -0.808866 which is FC = 0.57, not FC=0.56 as is stated in the paper. Also you should not call this a 0.44x reduction, which is a confusing presentation and just state in the paper that GJA1 RNA is reduced with FC = 0.57 FDR = 1.58E-93.

Reviewer #3 (Remarks to the Author):

The authors have addressed my concerns: their paper has very interesting findings, and it has been fun to contribute to the review process.

Authors' Response to Comments from the Editors and the Reviewers

In the following we list the comments from the reviewers in **bold** followed by our changes/explanations in plain text. Changes made in the manuscript are highlighted in blue.

REVIEWER COMMENTS

Reviewer #2 (Remarks to the Author):

The amendments have largely addressed my concerns. I would prefer the authors to simply present that GJA1 is down and not present it as two thresholds, particularly given that this is in iPS cells and the fold changes may differ in the disease relevant cell. However, the authors' presentation is sufficient and I would be happy to see this published, although three minor points are below:

Our response: We appreciate your kind and valuable suggestions for improving our paper.

Three minor comments:

- Why is the greyed zone in Fig 3c at +/- log₂ 0.8? This coincides with GJA1, but the paper and the legend states that its +/- log₂ 1? This looks like the threshold has been lowered to the FC of GJA1.

Our response: Our apologizes for the error. The greyed zone in Fig 3c has been revised to +/-1 in the revised version of Fig 3d, which is the threshold of the log₂(fold change).

- In general a heterozygous null does not imply a 50% reduction in expression. However, the paper contains a stronger statement than the reviewer's response. "However, a heterozygous null allele with a 50% decrease in expression is well tolerated in human beings since individuals carrying heterozygous null allele are unaffected according to a systematic review in our

previous study²⁷" I do believe that ref. 27 assesses expression and so that authors should highlight that the 50% decrease may accompany the heterozygous null.

Our response: Thanks, this has been revised accordingly.

- GJA1 has a log₂ FC of -0.808866 which is FC = 0.57, not FC=0.56 as is stated in the paper. Also you should not call this a 0.44x reduction, which is a confusing presentation and just state in the paper that GJA1 RNA is reduced with FC = 0.57 FDR = 1.58E-93.

Our response: Thanks a lot for your careful correction of our error. This has been revised accordingly.

Reviewer #3 (Remarks to the Author):

The authors have addressed my concerns: their paper has very interesting findings, and it has been fun to contribute to the review process.

Our response: Thanks a lot for your help and confirmation.